# When Lower-Order Terms Dominate: Adaptive Expert Algorithms for Heavy-Tailed Losses

**Antoine Moulin**[*]
Universitat Pompeu Fabra
antoine.moulin@upf.edu

**Emmanuel Esposito**[*]
Università degli Studi di Milano
emmanuel@emmanuelesposito.it

**Dirk van der Hoeven**
Leiden University
dirk@dirkvanderhoeven.com

## Abstract

We consider the problem setting of prediction with expert advice with possibly heavy-tailed losses, i.e. the only assumption on the losses is an upper bound on their second moments, denoted by $\theta$. We develop adaptive algorithms that do not require any prior knowledge about the range or the second moment of the losses. Existing adaptive algorithms have what is typically considered a lower-order term in their regret guarantees. We show that this lower-order term, which is often the maximum of the losses, can actually dominate the regret bound in our setting. Specifically, we show that even with small constant $\theta$, this lower-order term can scale as $\sqrt{KT}$, where $K$ is the number of experts and $T$ is the time horizon. We propose adaptive algorithms with improved regret bounds that avoid the dependence on such a lower-order term and guarantee $\mathcal{O}(\sqrt{\theta T \log(K)})$ regret in the worst case, and $\mathcal{O}(\theta \log(KT)/\Delta_{\min})$ regret when the losses are sampled i.i.d. from some fixed distribution, where $\Delta_{\min}$ is the difference between the mean losses of the second best expert and the best expert. Additionally, when the loss function is the squared loss, our algorithm also guarantees improved regret bounds over prior results.

## 1 Introduction

We study the problem of prediction with expert advice [Vovk, 1990, Littlestone and Warmuth, 1994], a sequential decision-making setting over $T$ rounds where each round $t$ goes as follows. The learner selects a probability distribution $p_t \in \mathcal{P} = \{p \in \mathbb{R}^K : p(i) \geq 0, \langle p_t, \mathbb{1} \rangle = 1\}$ over $K$ experts, suffers some loss $f_t(p_t) \in \mathbb{R}$, and subsequently observes the loss function $f_t \colon \mathcal{P} \to \mathbb{R}$. The goal in this problem, also known more briefly as the experts setting, is to control the (pseudo-)regret

$$R_T = \max_{i \in [K]} R_T(e_i) = \max_{i \in [K]} \mathbb{E}\left[\sum_{t=1}^{T} (f_t(p_t) - f_t(e_i))\right],$$

where the expectation is with respect to the randomness in $(f_t)_{t \in [T]}$, and $e_i \in \mathcal{P}$ is the $i$-th standard basis vector. The pseudo-regret measures the expected difference between the learner's cumulative loss and that of the best expert in hindsight. Throughout the paper, we assume that either the losses or the *outcomes* are heavy tailed, and provide a concrete answer to the concluding remarks of Mhammedi et al. [2019] that highlight the challenge of dealing with infrequent large values in the full-information setting. Such conditions arise in various practical settings involving noisy data,

---

[*]Equal contribution.

39th Conference on Neural Information Processing Systems (NeurIPS 2025).

outliers, or mechanisms like those in local differential privacy [Van der Hoeven, 2019]. Further motivations for our setting can be found in financial markets [Bradley and Taqqu, 2003], electricity forecasting [Li and Jones, 2019, Devaine et al., 2013], and predicting views on articles or videos (see, *e.g.*, Cha et al. [2007]).

A primary goal of this paper is to develop algorithms that achieve sub-linear regret without prior knowledge of a bound on the second moment of the losses, outcomes, or range of the losses $\max_{t \in [T], p \in \mathcal{P}} |f_t(p)|$. We will refer to such algorithms as *loss-range adaptive algorithms*. Crucially, unlike the related multi-armed bandit problem, where adaptivity to an unknown second moment is generally impossible without further assumptions [Genalti et al., 2024], we demonstrate this is achievable in the experts setting. Our second goal is to aim for algorithms that exhibit strong guarantees both in worst-case scenarios and in more benign stochastic environments, often referred to as *best-of-both-worlds* guarantees.

While several loss-range adaptive algorithms exist for linear losses, $f_t(p) = \langle p, \ell_t \rangle = \sum_{i=1}^{K} p(i)\ell_t(i)$ for some $\ell_t \in \mathbb{R}^K$, they are not designed to handle heavy-tailed losses [Blackwell, 1956, Cesa-Bianchi et al., 2007, De Rooij et al., 2014, Orabona and Pál, 2015, Mhammedi et al., 2019]. Standard algorithms achieve $\mathcal{O}(M\sqrt{T \log K})$ regret for losses bounded as $\max_{t,i} |\ell_t(i)| \leq M$. However, if the losses are drawn from some distribution supported on $\mathbb{R}$ and such that $\max_{t,i} \mathbb{E}[\ell_t(i)^2] \leq \theta$ for some $\theta > 0$, the regret guarantees of existing algorithms can degrade to $\mathcal{O}(\sqrt{KT})$, which is exponentially worse in $K$. The issue lies in what prior work consider a lower-order term, namely $\max_{t,i} |\ell_t(i)|$. Indeed, while this term is innocuous for bounded losses, we demonstrate that for heavy-tailed losses with second moment $\theta = \mathcal{O}(1)$, it can be as large as $\Omega(\sqrt{KT})$, thereby becoming the dominant factor in the regret bound.

This paper introduces adaptive algorithms that overcome this limitation. Specifically:

1. For linear losses, under Assumption 2.1, Algorithm 2 achieves $\mathcal{O}(\sqrt{\theta T \log(K)})$ worst-case regret, effectively removing the detrimental $\sqrt{K}$ dependency.

2. Under Assumption 2.1, Algorithm 1 achieves $\mathcal{O}(\sqrt{\theta T \log(KT)})$ worst-case regret, while also providing a $\mathcal{O}(\theta \log(KT)/\Delta_{\min})$ regret in self-bounded environments (Assumption 2.3), where $\Delta_{\min}$ is the gap in expected loss between the two best experts. This contrasts with prior algorithms whose guarantees would still be hampered by the $\mathcal{O}(\sqrt{KT})$ term.

3. For the squared loss, $f_t(p) = (\langle p, \mathbf{z}_t \rangle - y_t)^2$, where expert predictions $\mathbf{z}_{t,i}$ are bounded by $Y$ and the second moment of the outcomes $y_t$ is bounded by $\sigma$, Algorithm 3 achieves a regret of $\mathcal{O}((Y^2 + \sigma) \log(K))$, avoiding a $\mathcal{O}(T)$ regret that can arise for heavy-tailed outcomes.

The remainder of the paper is structured as follows: Section 2 presents the problem settings and our main results. Section 3 discusses related work in more detail. In Section 4, we establish why terms previously considered "lower-order" become dominant under heavy-tailed losses for existing algorithms. Section 5 presents our algorithms and sketches their regret analyses, highlighting the techniques used to circumvent the challenges posed by the lack of control over the range of losses. We conclude with a discussion of future work in Section 6.

## 2 Preliminaries and Results

We consider two settings. The first, sometimes referred to as the "Hedge setting" [Littlestone and Warmuth, 1994], can be seen as a special case of the experts setting [Vovk, 1990] with linear losses. In the second, we consider quadratic losses. Throughout the paper, we assume $K, T \geq 2$ without loss of generality.

### 2.1 Hedge Setting

In this section, each round $t = 1, \ldots, T$, goes as follows: the learner issues $p_t \in \mathcal{P}$, suffers loss $\langle p_t, \ell_t \rangle$, and then observes $\ell_t \in \mathbb{R}^K$. Let $\mathbb{E}_t[\cdot] = \mathbb{E}[\cdot | \mathcal{F}_t]$, where $\mathcal{F}_t$ is the sigma-field generated by the history $(\ell_1, \ldots, \ell_{t-1})$ up to round $t$. We will use the following assumption in the Hedge setting.

**Assumption 2.1** (finite second moments). There exists a scalar $\theta \in \mathbb{R}$ such that $\mathbb{E}_t[\ell_t(i)^2] \leq \theta$ for any $t \in [T]$ and any $i \in [K]$.

This assumption is more general than the standard bounded losses assumption: if $|\ell_t(i)| \leq M$ for all $i \in [K]$, $t \in [T]$, then Assumption 2.1 is satisfied with $\theta = M^2$. We *do not assume* to know $\theta$, which is the central algorithmic challenge in the Hedge setting. One of our main results is the following.

**Theorem 2.2.** *Consider the Hedge setting and suppose Assumption 2.1 holds. Then, there exists an algorithm that, without prior information on the losses, guarantees that* $R_T = \mathcal{O}\big(\sqrt{\theta T \log(K)}\big)$.

This is achieved by Algorithm 2, which we discuss later in Section 5.2. The proof can be found in Appendix C. While such a result seems expected, we found that loss-range adaptive algorithms in the literature can only guarantee $\mathcal{O}\big(\sqrt{\theta T \log(K)} + \sqrt{KT}\big)$ regret. We detail prior results in Section 3 and their issues in Section 4.

The following assumption allows us to obtain better regret bounds.

**Assumption 2.3.** There exist $\Delta_{\min}, C > 0$ and a unique $i^\star = \arg\min_{i \in [K]} \mathbb{E}\left[\sum_{t=1}^{T} \ell_t(i)\right]$ such that $R_T \geq \mathbb{E}\left[\sum_{t=1}^{T} \big(1 - p_t(i^\star)\big)\Delta_{\min}\right] - C$.

This is known as a self-bounded environment [Zimmert and Seldin, 2021] and has been studied in the Hedge setting by Amir et al. [2020]. An example setting in which Assumption 2.3 is satisfied is when the losses of the experts are sampled identically and independently from a fixed distribution, in which case $C = 0$. To see why, denote by $\Delta_i = \mathbb{E}[\ell_t(i)] - \mathbb{E}[\ell_t(i^\star)]$ and by $\Delta_{\min} = \min_{i \in [K]\backslash i^\star} \Delta_i$. We have that

$$R_T = \mathbb{E}\left[\sum_{t=1}^{T} \sum_{i \neq i^\star} p_t(i)\Delta_i\right] \geq \mathbb{E}\left[\sum_{t=1}^{T} \sum_{i \neq i^\star} p_t(i)\Delta_{\min}\right] = \mathbb{E}\left[\sum_{t=1}^{T} \big(1 - p_t(i^\star)\big)\Delta_{\min}\right]$$

and so Assumption 2.3 is satisfied. We have the following result under Assumption 2.3.

**Theorem 2.4.** *Consider the Hedge setting and suppose Assumption 2.1 holds. Then, there exists an algorithm that, without prior information on the losses, guarantees* $R_T = \mathcal{O}\big(\sqrt{\theta T \log(KT)}\big)$. *Furthermore, if Assumption 2.3 also holds, then the same algorithm simultaneously guarantees* $R_T = \mathcal{O}\Big(\frac{\theta \log(KT)}{\Delta_{\min}} + \sqrt{\frac{C\theta \log(KT)}{\Delta_{\min}}}\Big)$.

This is achieved by Algorithm 1, which we discuss in Section 5.1 together with a sketch of the proof. We provide a full proof in Appendix B.

## 2.2 Quadratic Losses

Here we introduce a second setting we consider. In each round $t \in [T]$, the learner observes the expert predictions $(\mathbf{z}_{t,i})_{i \in [K]}$, issues a prediction $\bar{\mathbf{z}}_t = \langle p_t, \mathbf{z}_t \rangle$, suffers the quadratic loss $f_t(p_t) = (\bar{\mathbf{z}}_t - y_t)^2$, and then observes $y_t$. We have the following result.

**Theorem 2.5.** *Consider quadratic losses and suppose that* $\max_{t,i} |\mathbf{z}_{t,i}| \leq Y$ *and* $\max_t \mathbb{E}_t\left[y_t^2\right] \leq \sigma$. *Then there exists an algorithm such that* $R_T = \mathcal{O}\left((Y^2 + \sigma)\log(KT)\right)$.

We prove this result in Appendix D. A sketch of the proof is provided in Section 5.3. Our analysis could also be extended to strongly convex losses. We leave this extension for future work.

**Notation.** For any $t \in [T]$, $i \in [K]$, let $r_t(i) = \langle p_t, \ell_t \rangle - \ell_t(i)$ be the instantaneous regret, $v_t(i) = r_t(i)^2$, and $\bar{v}_t = \sum_{i=1}^{K} p_t(i)v_t(i)$ be the variance of $\ell_t(I_t)$ with $I_t \sim p_t$. Furthermore, let $\bar{V}_T = \sum_{t=1}^{T} \bar{v}_t$, $V_T(i) = \sum_{t=1}^{T} v_t(i)$, $M_T = \mathbb{E}\left[\max_{t \in [T], i \in [K]} |r_t(i)|\right]$, and $L_T = \mathbb{E}\left[\max_{t \in [T], i \in [K]} |\ell_t(i)|\right]$. Finally, we denote by $i^\star \in \arg\min_{i \in [K]} \mathbb{E}\left[\sum_{t=1}^{T} \ell_t(i)\right]$ the best expert in hindsight.

# 3 Related Work

There are several loss-range adaptive algorithms in the Hedge setting. A summary can be found in Table 1. Perhaps the most well-known loss-range adaptive algorithm is the exponential weights

| | Assumption 2.1 | Assumptions 2.1 and 2.3 with $C = 0$ |
|---|---|---|
| Cesa-Bianchi et al. [2007]; De Rooij et al. [2014] | $\sqrt{\theta T \log(K)} + M_T$ | $\theta \log(K)\Delta_{\min}^{-1} + M_T$ |
| Mhammedi et al. [2019] | $\sqrt{\theta T \log(K)} + M_T$ | $\theta \log(K)\Delta_{\min}^{-1} + M_T$ |
| Algorithm 1 (**Ours**) | $\sqrt{\theta T \log(KT)}$ | $\theta \log(KT)\Delta_{\min}^{-1}$ |
| Algorithm 2 (**Ours**) | $\sqrt{\theta T \log(K)}$ | $\sqrt{\theta T \log(K)}$ |

Table 1: An overview of the most relevant loss-range adaptive algorithms in the literature, ignoring constants. Some of the results in the middle column follow from an application of Jensen's inequality.

algorithm [Vovk, 1990, Littlestone and Warmuth, 1994, Cesa-Bianchi et al., 1997] combined with the doubling trick [Auer et al., 1995]. This combination leads to a $\mathcal{O}\left(L_T\sqrt{T\log(K)}\right)$ regret bound. Cesa-Bianchi et al. [2007] provide a refined version of the exponential weights algorithm combined with a refined doubling trick to obtain a $\mathcal{O}\left(\mathbb{E}\left[\sqrt{\overline{V}_T \log(K)}\right] + M_T \log(K)\right)$ regret bound. De Rooij et al. [2014] also prove the same regret bound for an algorithm based on the exponential weights algorithm, with the added benefit that their algorithm does not use restarts to achieve this result. Orabona and Pál [2015] provide a generic analysis of follow the regularized leader and online mirror descent that, under mild assumptions on the regularizer, leads to loss-range adaptive algorithms which guarantee that the regret is at most $\mathcal{O}\left(\mathbb{E}\left[\sqrt{\sum_t \max_i \ell_t(i)^2}\right]\right)$, but this can be trivially seen to be at least $M_T$. Mhammedi et al. [2019] provide a loss-range adaptive version of Squint [Koolen and Van Erven, 2015], which guarantees a regret bound scaling as $\mathcal{O}\left(\mathbb{E}\left[\sqrt{V_T(i^\star)\log(K)}\right] + M_T \log(K)\right)$. Flaspohler et al. [2021] show that regret matching [Blackwell, 1956, Hart and Mas-Colell, 2000] and regret matching+ [Tammelin et al., 2015] are both loss-range adaptive algorithm, but unfortunately with unsatisfactory regret bounds. Under Assumption 2.1, Wintenberger [2024] provides a loss-range adaptive algorithm with $\mathcal{O}\left(\log(K)\sqrt{\theta T} + \mathbb{E}\left[\sum_{i=1}^K \log\left(1 + \frac{\max_t |r_t(i)|}{m_{t,i}}\right)\right]\right)$ regret, where $m_{t,i}$ is the first non-null observation of $|r_t(i)|$. Unfortunately, it is not clear how to control $\frac{\max_t |r_t(i)|}{m_{t,i}}$ as $m_{t,i}$ can be made arbitrarily close to 0 by an adversary.

Orseau and Hutter [2024], Cutkosky [2019] provide generic templates to make most Hedge algorithms loss-range adaptive, at the cost of a $M_T$ term in the regret bound. Orseau and Hutter [2024] make use of the following observation: if the algorithm makes too extreme of an update it might never recover. Therefore, it is sometimes better to ignore certain losses. Cutkosky [2019] observed that one can always guarantee that the algorithm does not make such an extreme update by feeding the algorithm slightly shrunken losses, a technique also employed by Mhammedi et al. [2019], Mhammedi [2022]. Specifically, they feed the algorithm the losses

$$\widetilde{\ell}_t(i) = \ell_t(i)\min\left\{1, \frac{\max_{s\in[t-1],j\in[K]}|\ell_s(j)|}{\max_{j\in[K]}|\ell_t(j)|}\right\} = \ell_t(i) \cdot \frac{\max_{s\in[t-1],j\in[K]}|\ell_s(j)|}{\max_{s\in[t],j\in[K]}|\ell_s(j)|}.$$

This ensures the range of the losses we feed to the algorithm is known before choosing $p_t$. The cost for using $\widetilde{\ell}_t$ rather than $\ell_t$ in the algorithm is minor at first sight:

$$\sum_{t=1}^T \sum_{i=1}^K p_t(i)\big(\ell_t(i) - \widetilde{\ell}_t(i)\big) = \sum_{t=1}^T \sum_{i=1}^K p_t(i)\ell_t(i)\left(1 - \frac{\max_{s\in[t-1],j\in[K]}|\ell_s(j)|}{\max_{s\in[t],j\in[K]}|\ell_s(j)|}\right)$$

$$\leq \sum_{t=1}^T \max_{s'\in[t],i\in[K]}|\ell_{s'}(i)|\left(1 - \frac{\max_{s\in[t-1],j\in[K]}|\ell_s(j)|}{\max_{s\in[t],j\in[K]}|\ell_s(j)|}\right) = \max_{t\in[T],i\in[K]}|\ell_t(i)|.$$

However, this term can be prohibitively large as we will show in Section 4. Instead, we adapt and combine the ideas of Cutkosky [2019], Orseau and Hutter [2024]. We develop a coordinate-wise version of the clipping technique of Cutkosky [2019]. Unfortunately, this is not sufficient for our needs and we need to combine it with a coordinate-wise version of the null-updates of Orseau and Hutter [2024] and the multi-scale entropic regularizer of Bubeck et al. [2019] to guarantee a satisfactory regret bound. Gökcesu and Kozat [2022] also claim to provide loss-range adaptive algorithms, but two of their results seem to contain mistakes. We provide details in Appendix E.

**Scale-free algorithms.** A related but different objective is obtaining scale-free or equivalently scale-invariant algorithms. An algorithm is said to be scale-free if the predictions of the algorithm do not change if the sequence of losses is multiplied by a positive constant. While scale-free algorithms are loss-range adaptive the converse is not necessarily true. Mhammedi [2022] provide a generic wrapper to make any algorithm scale-free, under some mild assumptions on the algorithm. However, this comes at the cost of an additive $M_T$. The algorithms of De Rooij et al. [2014], Orabona and Pál [2015] are known to be scale-free. Unfortunately, it is not clear whether our algorithms are also scale-free.

**Adaptive algorithms for bounded losses.** If we assume that losses are bounded, *e.g.*, $\ell_t(i) \in [0, 1]$, then there are several works that provide best-of-both-worlds results. Gaillard et al. [2014], Koolen et al. [2016] show that so-called second-order bounds simultaneously guarantee $\mathcal{O}\big(\sqrt{T \log(K)}\big)$ regret without further assumptions on the loss and $\mathcal{O}(\log(K)/\Delta_{\min})$ regret under Assumption 2.3 with $C = 0$. This also implies that the results of Cesa-Bianchi et al. [2007], De Rooij et al. [2014], Koolen and Van Erven [2015], Chen et al. [2021] all lead to small regret under Assumption 2.3 with $C = 0$ while also being robust to more difficult environments. Mourtada and Gaïffas [2019] show that the exponential weights algorithm with a decreasing learning rate guarantees $\mathcal{O}(\log(K)/\Delta_{\min})$ regret under Assumption 2.3 with $C = 1$ while simultaneously guaranteeing $\mathcal{O}\big(\sqrt{T \log(K)}\big)$ regret in the worst case. The only work that we are aware of that treats Assumption 2.3 with $C > 0$ is by Amir et al. [2020], who show that the exponential weights algorithm with a decreasing learning rate guarantees $\mathcal{O}(\log(K)/\Delta_{\min} + C)$ regret.

**Online learning with heavy-tailed losses.** To the best of our knowledge, we are the first to consider heavy tailed losses in the expert setting. A related setting is studied by Bubeck et al. [2013], who introduce the multi-armed bandits with heavy tails setting. The main difference with our setting is that the learner only gets to see the loss for the chosen action each round, which significantly reduces the amount of available feedback to the learner. Remarkably, without further assumptions, one cannot adapt to $\theta$, as shown by Genalti et al. [2024]. However, under some additional assumptions on the loss, it is possible to adapt to $\theta$ with bandit feedback [Lee et al., 2020, Ashutosh et al., 2021, Huang et al., 2022, Genalti et al., 2024, Chen et al., 2025].

## 4 Lower-Order Terms with Unbounded Losses

In this section, we show that what have been considered lower-order terms in the literature may actually dominate regret bounds.

### 4.1 Lower-Order Terms in the Hedge Setting

We provide three results that imply that if a regret bound contains $M_T$ or $L_T$, then this "lower-order term" can dominate the regret bound. We rely on the following observation, proved in Appendix A.

**Lemma 4.1.** *Fix any $n \in \mathbb{N} \setminus \{0\}$. Let $X_1, \ldots, X_n$ be non-negative i.i.d. random variables such that*

$$X_j = \begin{cases} 0 & w.p. \ 1 - \frac{1}{n} \\ \sqrt{n} & w.p. \ \frac{1}{n} \end{cases} .$$

*for each $j \in [n]$. Then, $\mathbb{E}[X_j^2] = 1$ for any $j \in [n]$ and $\frac{1}{2}\sqrt{n} \leq \mathbb{E}[\max\{X_1, \ldots, X_n\}] \leq \sqrt{n}$.*

While the distribution in Lemma 4.1 might appear unnatural, the Fréchet distribution, which is often used in economics, satisfies similar properties as the distribution of Lemma 4.1; see Lemma A.1 in Appendix A. Lemma 4.1 immediately leads to the following result.

**Proposition 4.2.** *There exists a distribution for $\ell_1, \ldots, \ell_T$ that satisfies Assumption 2.1 with $\theta = 4$ such that $L_T \geq \frac{1}{2}\sqrt{KT}$.*

*Proof.* Simply choose $\ell_t(i) = \mu_{t,i} + \xi_{t,i}\varepsilon_{t,i}$ where $\mu_{t,i} \in [0, 1]$ is chosen arbitrarily, $\xi_{t,i}$ is a Rademacher random variable, and $\varepsilon_{t,i}$ follow the distribution specified in Lemma 4.1 with $n = KT$. Then, Lemma 4.1 provides the result after using $\mathbb{E}_t[\ell_t(i)^2] \leq 2\mu_{t,i}^2 + 2\mathbb{E}_t[\varepsilon_{t,i}^2] \leq 4$. $\qquad\square$

In the worst case, Proposition 4.2 implies that any algorithm with a regret bound of the form $R_T \leq \sqrt{\theta T \log(K)} + L_T$ will be dominated by $L_T$ for large enough $K$. Existing algorithms sometimes have a $M_T \leq 2L_T$ term in the regret instead. It is natural to question whether $M_T$ can be small enough. The following proposition shows that $M_T$, like $L_T$, can be prohibitively large.

**Proposition 4.3.** *Suppose one can guarantee a bound $R_T \leq B_T^\star + M_T$, with $B_T^\star \geq 0$. There exists a distribution with $\theta = 3$ such that $B_T^\star + M_T \geq B_T^\star + \frac{1}{16}\sqrt{KT} - 1$.*

The proof of Proposition 4.3 can be found in Appendix A. While Proposition 4.3 shows that any regret bound with a term $M_T$ is $\mathcal{O}(\sqrt{KT})$, Theorem 2.2 guarantees that Algorithm 2 achieves $R_T = \mathcal{O}(\sqrt{\theta T \log(K)})$. For loss sequences that satisfy Assumption 2.3, the situation is even more dire, as one then hopes to guarantee $\mathcal{O}(\theta \log(K)/\Delta_{\min})$ regret. Proposition 4.3 shows that if one has a regret bound of the form $\theta \log(K)/\Delta_{\min} + M_T$, in the worst case this bound is $\mathcal{O}(\sqrt{KT})$. In contrast, in Theorem 2.4 we provide a $\mathcal{O}(\theta \log(KT)/\Delta_{\min})$ regret bound, which is exponentially better in $K$ and $T$. Since these results only affect upper bounds on the regret, one can ask whether existing algorithms could benefit from a more careful analysis. We do not think this is possible for existing algorithms and sketch an argument in Appendix A. Furthermore, to illustrate their failure modes, we compare empirically those algorithms with the ones we introduce in Section 5 in Appendix F.

## 4.2 Lower-Order Terms for the Squared Loss

Since the squared loss is exp-concave (or mixable), an appealing approach is to use exponential weights with a suitable learning rate to obtain a regret bound that is seemingly independent of $T$. Indeed, a simple calculation shows that the function $g_t(\cdot) = (y_t - \cdot)^2$ is $(2 \max_{t,i}(\mathbf{z}_{t,i} - y_t)^2)^{-1}$-exp-concave. If one knows $\max_{t,i}(\mathbf{z}_{t,i} - y_t)^2$, this would lead to the following bound [Vovk, 1990]:

$$R_T \leq 2 \log(K) \cdot \mathbb{E}\left[\max_{t,i}\,(\mathbf{z}_{t,i} - y_t)^2\right] \tag{1}$$

However, the $\max_{t,i}(\mathbf{z}_{t,i} - y_t)^2$ factor might lead to a poor guarantee. Suppose that $|\mathbf{z}_{t,i}| \leq Y$ and that $\mathbb{E}\left[y_t^2\right] \leq \theta < \infty$. If $y_t = \mu_t + \xi_t \varepsilon_t$, where $|\mu_t| \leq 1$, $\xi_t$ is a Rademacher random variable, and $\varepsilon_t$ follows the distribution specified in Lemma 4.1 with $n = T$, then

$$\mathbb{E}\left[\max_{t,i}\,(\mathbf{z}_{t,i} - y_t)^2\right] \geq \mathbb{E}\left[\max_t y_t^2 - 2Y|y_t|\right] \geq \mathbb{E}\left[\max_t |y_t|\right]^2 - 2Y \mathbb{E}\left[\max_t |y_t|\right]$$

$$\geq \mathbb{E}\left[\max_t |\varepsilon_t|\right]^2 - 2(Y+1)\left(1 + \mathbb{E}\left[\max_t |\varepsilon_t|\right]\right) \geq \frac{1}{2}T - 2(Y+1)\sqrt{T}\,,$$

which can be excessively large. Even more so, one would need to know $\max_{t,i}(\mathbf{z}_{t,i} - y_t)^2$ to obtain the regret bound in Equation (1). The results of Wintenberger [2017] (see also Van der Hoeven et al. [2022]) imply that the algorithm of Mhammedi et al. [2019], designed for the Hedge setting, applied to the losses $\ell_t(i) = 2\mathbf{z}_{t,i}(\bar{\mathbf{z}}_t - y_t)$ would lead to a bound of order

$$R_T = \mathcal{O}\left(\left(Y^2 + \theta + \mathbb{E}\left[\max_t |\bar{\mathbf{z}}_t - y_t|\right]\right)\log(K)\right)\,.$$

The $\mathbb{E}[\max_t |\bar{\mathbf{z}}_t - y_t|]$ term can also be problematic. Indeed, if $y_t = \mu_t + \xi_t \varepsilon_t$, where $\mu_t \in [-1, 1]$, $\xi_t$ is a Rademacher random variable, and $\varepsilon_t$ follows the distribution specified by in Lemma 4.1 with $n = KT$, then $\mathbb{E}\left[\max_t |\bar{\mathbf{z}}_t - y_t|\right] \geq \mathbb{E}\left[\max_t |\bar{\mathbf{z}}_t|\right] - Y = \frac{1}{2}\sqrt{T} - Y$. Thus, if $Y \geq 2$ then $Y^2 + \theta + \mathbb{E}\left[\max_t |\bar{\mathbf{z}}_t - y_t|\right] \geq \frac{1}{2}(Y^2 + \sqrt{T})$. On the other hand, for the same $y_t$, Theorem 2.5 implies that Algorithm 3 guarantees a regret bound of order $(4 + Y^2)\log(KT)$.

## 5 Algorithms

In this section, we present two types of algorithms and provide a sketch for their regret analysis. The first algorithm, called `LoOT-Free OMD` (Lower-Order Term Free), is an instance of Online Mirror Descent (OMD), whereas the second, called `LoOT-Free FTRL`, is an instance of Follow The Regularized Leader (FTRL). Both algorithms are run on clipped versions of the instantaneous regrets,

---

**Algorithm 1** `LoOT-Free OMD`

---

**Inputs:** Number of experts $K \geq 2$, truncation parameter $\alpha \in (0, 1]$, learning rate $\beta > 0$.
**Initialize:** $p_1(i) \leftarrow 1/K$ for all $i \in [K]$.
**for** $t = 1, \ldots, T$ **do**
    Predict $p_t$, incur loss $\langle p_t, \ell_t \rangle$ and observe $\ell_t$.
    Set $r_t(i) \leftarrow \langle p_t, \ell_t \rangle - \ell_t(i)$ for all $i \in [K]$.
    Set $v_t(i) \leftarrow r_t(i)^2$ for all $i \in [K]$, and $\bar{v}_t \leftarrow \sum_{i=1}^{K} p_t(i) v_t(i)$.
    **if** $\sum_{s \leq t} \bar{v}_s > 0$ **then**
        Set $\eta_{t,i} \leftarrow \beta \max\left\{\sum_{s \leq t} \bar{v}_s, \sum_{s \leq t} v_s(i)\right\}^{-1/2}$ for all $i \in [K]$.
        Set $\widetilde{\ell}_t(i) \leftarrow -r_t(i) \mathbb{1}\left(|r_t(i)| \leq 1/\eta_{t,i}\right)$ for all $i \in [K]$.
        Set $p_{t+1} \leftarrow \arg\min_{p \in \mathcal{P}_\alpha} \langle p, \widetilde{\ell}_t \rangle + D_t(p \| p_t)$.
    **end if**
**end for**

---

but one could obtain similar worst-case guarantees by running the algorithms on clipped versions of the losses.

Before we describe the algorithms in more detail, we first need some definitions. For any $t \in [T]$, let $\eta_t \in \mathbb{R}_{>0}^K$ be a vector of learning rates and $\psi_t : x \in \mathbb{R}_{\geq 0}^K \mapsto \langle \mathrm{Diag}\left(\eta_t^{-1}\right) x, \log x \rangle$ be the regularizer, where $y^{-1}$ is the coordinate-wise inverse, $\log y$ is the coordinate-wise logarithm, and $\mathrm{Diag}(y)$ is the diagonal matrix with diagonal entries $y_1, \ldots, y_K$ for any $y \in \mathbb{R}^K$. Note that, for any $x \in \mathbb{R}_{\geq 0}^K$, the gradient of $\psi_t$ is given by $\nabla \psi_t(x) = \mathrm{Diag}\left(\eta_t^{-1}\right) \log x + \eta_t^{-1}$. Consider also its Bregman divergence, defined for any $x \in \mathbb{R}_{\geq 0}^K$ and $y \in \mathbb{R}_{>0}^K$ as $D_t(x \| y) = \psi_t(x) - \psi_t(y) - \langle \nabla \psi_t(y), x - y \rangle = \sum_{i=1}^{K} \frac{1}{\eta_{t,i}} \left[ x_i \log\left(\frac{x_i}{y_i}\right) - x_i + y_i \right]$. Finally, denote by $\mathcal{P}_\alpha = \{p \in \mathcal{P} : \min_i p_i \geq \alpha/K\}$ the probability simplex truncated by $\alpha/K$, for any $\alpha \in [0, 1]$.

### 5.1 OMD-based Algorithm

The main challenge in designing an algorithm for losses for which you do not know the range comes from proving the stability of the algorithm, which is to say that $p_t(i) \approx p_{t+1}(i)$. An analysis based on strong convexity circumvents this challenge. Orabona and Pál [2015] show that if the regularizer $\phi$ is strongly convex with respect to some norm $\| \cdot \|$, the regret can be $\mathcal{O}\left(\sqrt{\max_{p \in \mathcal{P}} \phi(p) \sum_{t=1}^{T} \|\ell_t\|_\star^2}\right)$, where $\| \cdot \|_\star$ is the dual norm. However, if one uses the (shifted) negative Shannon entropy as $\phi$, this will lead to a $\mathcal{O}\left(\sqrt{\log(K) \sum_{t=1}^{T} \|\ell_t\|_\infty^2}\right)$ bound, which is $\mathcal{O}(\sqrt{KT})$ in the worst case. A more careful analysis that avoids strong convexity arguments can be found in De Rooij et al. [2014], but unfortunately this analysis does not avoid a problematic lower-order term. To avoid such issues, we make use of the multi-scale entropic regularizer of Bubeck et al. [2019]. If the range of the losses is known a-priori, Bubeck et al. [2019] show that for bounded losses and known $\max_t |\ell_t(i)|$ the regret against expert $i$ then scales as $\mathbb{E}[\max_t |\ell_t(i)|]\sqrt{T \log(K)}$. However, this alone is not sufficient, as it is not clear how to prove that the algorithm is stable without an a-priori uniform bound on the losses, nor is such a regret bound sufficiently small. Instead, we carefully clip the losses when they are excessively large. Combining these ideas leads to the following guarantee for Algorithm 1 with $\alpha = \frac{1}{T}$ and $\beta = \sqrt{\log(KT)}$:

$$R_T(e_{i^\star}) = \sum_{t=1}^{T} r_t(i^\star) = \mathcal{O}\left(\sqrt{\log(KT)\left(\bar{V}_T + V_T(i^\star)\right)}\right). \tag{2}$$

Notice that the above upper bound holds with probability one for any sequence of losses and, in turn, leads to the guarantees of Theorem 2.4. A full proof of the above result can be found in Appendix B. Here we provide some intuition.

*Proof sketch of Theorem 2.4.* First, observe that Algorithm 1 is an instance of OMD adopting the time-varying regularizer $\psi_t$ with some additional tricks. We only update $p_t$ if $\sum_{s \leq t} \bar{v}_s > 0$. The only

case where this is not true is if in all rounds up to and including round $t$, $r_t(i) = 0$ for all $i \in [K]$, in which case we can simply ignore these rounds in the analysis. From standard OMD analysis (see for example Orabona [2025]), we know that if we run OMD on losses $\widetilde{\ell}_t$, for any fixed $j \in [K]$,

$$\sum_{t=1}^{T} \left( \langle p_t, \widetilde{\ell}_t \rangle - \widetilde{\ell}_t(j) \right) = \mathcal{O}\left( \frac{\log(K/\alpha)}{\eta_{T,j}} + \sum_{t=1}^{T} \sum_{i=1}^{K} \eta_{t,i} \widetilde{p}_t(i) \widetilde{\ell}_t(i)^2 \right),$$

where $\widetilde{p}_t(i) = p_t(i) \exp\left( -\eta_{t,i} \widetilde{\ell}_t(i) \right)$. At this point, we face our main challenge. We would like to show that $\widetilde{p}_t(i) \approx p_t(i)$. To do so, we would need to show that $|\eta_{t,i} \widetilde{\ell}_t(i)| \leq 1$. We force this to be true by simply setting $\widetilde{\ell}_t(i) = 0$ if $|r_t(i)| > \frac{1}{\eta_{t,i}}$ and $\widetilde{\ell}_t(i) = -r_t(i)$ otherwise. Of course, there is a price to pay for clipping the losses, but this price is negligible:

$$\left| \widetilde{\ell}_t(i) - (-r_t(i)) \right| = |r_t(i)| \mathbb{1}\left( |r_t(i)| > \frac{1}{\eta_{t,i}} \right) = \frac{r_t(i)^2}{|r_t(i)|} \mathbb{1}\left( |r_t(i)| > \frac{1}{\eta_{t,i}} \right) \leq \eta_{t,i} v_t(i) . \quad (3)$$

Thus, after noting that $\ell_t$ and $-r_t$ only differ from a constant, we have that the regret against expert $j$ is $\sum_{t=1}^{T} \left( \langle p_t, \ell_t \rangle - \ell_t(j) \right) = \sum_{t=1}^{T} \left( \langle p_t, -r_t \rangle - (-r_t(j)) \right)$, and we can replace $-r_t$ by the truncated losses $\widetilde{\ell}_t$ by paying the cost shown in Equation (3) and use the guarantee from OMD

$$\sum_{t=1}^{T} \left( \langle p_t, \ell_t \rangle - \ell_t(j) \right) = \mathcal{O}\left( \sum_{t=1}^{T} \left( \langle p_t, \widetilde{\ell}_t \rangle - \widetilde{\ell}_t(j) \right) + \sum_{t=1}^{T} \left( \eta_{t,j} v_t(j) + \sum_{i=1}^{K} p_t(i) \eta_{t,i} v_t(i) \right) \right)$$

$$= \mathcal{O}\left( \frac{\log(K/\alpha)}{\eta_{T,j}} + \sum_{t=1}^{T} \left( \eta_{t,j} v_t(j) + \sum_{i=1}^{K} p_t(i) \eta_{t,i} v_t(i) \right) \right) .$$

After controlling the sum by using the definition of $\eta_{t,i}$, the bound in Equation (2) follows as

$$\sum_{t=1}^{T} \left( \eta_{t,j} v_t(j) + \sum_{i=1}^{K} p_t(i) \eta_{t,i} v_t(i) \right) \leq \beta \sum_{t=1}^{T} \left( \frac{v_t(j)}{\sqrt{V_t(j)}} + \frac{\bar{v}_t}{\sqrt{\bar{V}_t}} \right) \leq 4\beta \sqrt{\bar{V}_T + V_T(j)} ,$$

where the first inequality follows from the definition of $\eta_{t,i}$ and the second follows from, for example, Lemma 4.13 in Orabona [2025]. $\qquad \square$

Regarding computational aspects, note the optimization problem defining $p_{t+1}$ is done on a truncated simplex, thus it does not have a closed-form solution but can be computed efficiently with a line-search as done by Chen et al. [2021].

## 5.2 FTRL-based Algorithm

Taking some ideas from the previous OMD-based algorithm, we derive an instantiation of FTRL described by Algorithm 2. For FTRL we use once more the multi-scale entropic regularizer $\psi_t$ and a similar clipping of the losses to resolve the same issues that follow from the lack of any prior knowledge of the loss range. However, the FTRL framework has some fundamental differences compared to OMD, which in turn require further adjustments. To understand these differences, we will sketch the proof of Theorem 2.2 below, whose result is obtained by Algorithm 2 with $\beta = \sqrt{\log(K)}$. The detailed proof of Theorem 2.2 can be found in Appendix C. Here we provide the main ideas. We actually prove a stronger regret guarantee than provided in Theorem 2.2:

$$R_T(e_{i^\star}) = \sum_{t=1}^{T} r_t(i^\star) = \mathcal{O}\left( \sqrt{\log(K)\left( \bar{V}_T + V_T(i^\star) \right)} + \frac{1}{K} \sum_{i=1}^{K} \sqrt{V_T(i)} \right) , \quad (4)$$

which then implies the statement of Theorem 2.2.

To prove this regret bound we make use of the standard FTRL analysis (e.g., see Orabona [2025]) with some additional tricks. Note that, differently from Algorithm 1, we replace the multi-scale entropic regularizer $\psi_t$ with its Bregman divergence. This is to ensure the monotonicity of the regularizer, i.e., $\varphi_{t+1}(x) \geq \varphi_t(x)$ for all $x \in \mathcal{P}$. This monotonicity is necessary for the proof of the FTRL regret bound. The fact that we use the Bregman divergence generated by $\psi_t$ rather than $\psi_t$ directly

---

**Algorithm 2** `LoOT-Free FTRL`

---

**Inputs:** Number of experts $K \geq 2$, learning rate coefficient $\beta > 0$.
**Initialize:** $p_1(i) \leftarrow 1/K$ and $b_0(i) \leftarrow 0$ for all $i \in [K]$.
**for** $t = 1, \ldots, T$ **do**
    Predict $p_t$, incur loss $\langle p_t, \ell_t \rangle$ and observe $\ell_t$.
    Set $r_t(i) \leftarrow \langle p_t, \ell_t \rangle - \ell_t(i)$ for all $i \in [K]$.
    Set $v_t(i) \leftarrow r_t(i)^2$ for all $i \in [K]$.
    Set $\bar{v}_t \leftarrow \sum_{i=1}^{K} p_t(i) v_t(i)$.
    **if** $\sum_{s \leq t} \bar{v}_s > 0$ **then**
        Set $b_t(i) \leftarrow \max\left\{ \sum_{s \leq t} \bar{v}_s, \sum_{s \leq t} v_s(i) \right\}^{1/2}$ for all $i \in [K]$.
        Set $\eta_{t,i} \leftarrow \beta / b_t(i)$ for all $i \in [K]$.
        Set $\widetilde{\ell}_t(i) \leftarrow -r_t(i) \frac{b_{t-1}(i)}{b_t(i)} \mathbb{1}\left( |r_t(i)| \leq 1/\eta_{t,i} \right)$ for all $i \in [K]$.
        Set $p_{t+1} \leftarrow \arg\min_{p \in \mathcal{P}} \sum_{s \leq t} \langle p, \widetilde{\ell}_s \rangle + D_t(p \| p_1)$.
    **end if**
**end for**

---

as the regularizer also allows us to avoid the $\log(T)$ term of the OMD regret bound, at the cost of a $\frac{1}{K} \sum_{i=1}^{K} \sqrt{\sum_{t=1}^{T} v_t(i)}$ term in the regret. While this term is prohibitively big when we try to obtain improved regret bounds under Assumption 2.3, in expectation this term scales as $\mathcal{O}(\sqrt{\theta T})$ under Assumption 2.1, thus preserving the desired final bound.

*Proof sketch of Theorem 2.2.* Since the $p_t$ in Algorithm 2 comes from an instance of FTRL over the losses $\widetilde{\ell}_t$, we can apply a standard FTRL bound (for example, Lemma 7.16 of Orabona [2025]) to see that, for any fixed $j \in [K]$,

$$\sum_{t=1}^{T} \left( \langle p_t, \widetilde{\ell}_t \rangle - \widetilde{\ell}_t(j) \right) = \mathcal{O}\left( \frac{\log K}{\eta_{T,j}} + \frac{1}{K} \sum_{i=1}^{K} \frac{1}{\eta_{t,i}} + \sum_{t=1}^{T} \sum_{i=1}^{K} \eta_{t-1,i} \widetilde{p}_t(i) \widetilde{\ell}_t(i)^2 \right) , \qquad (5)$$

where $\widetilde{p}_t(i) = p_t(i) \exp\left( -\eta_{t-1,i} \widetilde{\ell}_t(i) \right)$. Here we reach the main challenge of adapting FTRL to our setting. The third term illustrates crucial differences compared to OMD: the learning rate $\eta_{t-1,i}$ instead of $\eta_{t,i}$ appears as a multiplicative factor and in the definition of $\widetilde{p}_t$. We resolve this issue by rescaling the loss $-r_t(i)$ by a factor $b_{t-1}(i)/b_t(i)$ in the definition of $\widetilde{\ell}_t(i)$. Still, this rescaled loss can be too big, which is why we also clip the loss when necessary. Specifically, the clipping with a rescaling factor ensures, by the definitions of $b_t(i)$ and $\eta_{t,i}$, that $\eta_{t-1,i} \widetilde{\ell}_t(i)^2 \leq \frac{b_{t-1}(i)}{b_t(i)} \eta_{t-1,i} r_t(i)^2 = \eta_{t,i} v_t(i)$ and, similarly, that $|\eta_{t-1,i} \widetilde{\ell}_t(i)| \leq 1$. Thus, we can bound the third term on the right hand side of Equation (5) by $\mathcal{O}\left( \sum_{t=1}^{T} \sum_{i=1}^{K} \eta_{t,i} p_t(i) v_t(i) \right)$, which we already know how to control from the analysis of Algorithm 1.

Another consequence of the new definition of $\widetilde{\ell}_t$ is that it increases the cost of clipping the losses. In comparison to Equation (3), it now additionally presents terms

$$|r_t(i)| \left( 1 - \frac{b_{t-1}(i)}{b_t(i)} \right) \mathbb{1}\left( |r_t(i)| \leq \frac{1}{\eta_{t,i}} \right) \leq \frac{1}{\eta_{t,i}} - \frac{1}{\eta_{t-1,i}}$$

but the sums involving them are also nicely behaved. Combining all these observations leads to Equation (4). $\qquad \square$

While the optimization problem suggests $p_t$ is given by a softmax function, the coordinate-dependent learning rate prevents us from computing the normalization constant in closed-form (see Appendix C.1 for details). However, we can still compute it efficiently with a line-search as in Algorithm 1 from Bubeck et al. [2019].

## 5.3 Algorithm for the Squared Loss

The algorithm we use for the squared loss, Algorithm 3 can be found in Appendix D. It is exactly the same as Algorithm 1, but then run on the losses

$$\ell_t(i) = \mathbf{z}_{t,i}(\bar{\mathbf{z}}_t - y_t) + \frac{1}{2}(\mathbf{z}_{t,i} - y_t)^2 \ ,$$

where $\bar{\mathbf{z}}_t = \langle p_t, \mathbf{z} \rangle$. The inspiration from this surrogate loss comes from two inequalities for the squared loss:

$$(a - y)^2 - (b - y)^2 = 2(y - a)(b - a) - (a - b)^2 \ ,$$

$$\left(\frac{a+b}{2} - y\right)^2 \leq \frac{(a-y)^2}{2} + \frac{(b-y)^2}{2} - \frac{(a-b)^2}{8} \ .$$

By carefully applying these inequalities we find that, for any fixed $j \in [K]$,

$$\sum_{t=1}^{T}\left((\bar{\mathbf{z}}_t - y_t)^2 - (\mathbf{z}_{t,j} - y_t)^2\right)$$

$$\leq \sum_{t=1}^{T}\left(\sum_{i=1}^{K} p_t(i)\ell_t(i) - \ell_t(j) - \sum_{i=1}^{K} p_t(i)\frac{(\mathbf{z}_{t,i} - \bar{\mathbf{z}}_t)^2}{8} - \frac{(\mathbf{z}_{t,j} - \bar{\mathbf{z}}_t)^2}{2}\right).$$

The negative quadratics on the right-hand-side of the equation above allow us prove the regret bound of Theorem 2.5. We show that, for any fixed $j \in [K]$,

$$R_T(e_j) = \mathcal{O}\left(\mathbb{E}\left[\sqrt{\log(KT)(\bar{V}_T + V_T(j))}\right] - \mathbb{E}\left[\sum_{t=1}^{T}\left(\sum_{i=1}^{K}\frac{p_t(i)(\mathbf{z}_{t,i} - \bar{\mathbf{z}}_t)^2}{8} + \frac{(\mathbf{z}_{t,j} - \bar{\mathbf{z}}_t)^2}{2}\right)\right]\right)$$

$$= \mathcal{O}\left((Y^2 + \sigma)\log(KT)\right) \ .$$

A detailed proof can be found in Appendix D.

## 6 Future Work

One direction to explore is whether the $\log(T)$ factor in Theorem 2.4 is necessary to obtain best-of-both-worlds guarantees. Potentially, an improved analysis of the FTRL algorithm will do the trick, as we know that OMD can be inferior to FTRL [Amir et al., 2020]. While we understand how to obtain best-of-both-worlds guarantees for FTRL with bounded losses [Mourtada and Gaïffas, 2019, Amir et al., 2020], it is unclear how to adapt these analyses to our setting. Another interesting direction is to see whether the ideas in Section 5.3 can be extended to strongly convex and exp-concave losses. We believe that the former is relatively straightforward, but the latter might be highly challenging. Another relatively straightforward extension could be to adapt to any moment of the loss, *i.e.*, adapt to $\mathbb{E}_t[|\ell_t(i)|^\alpha]$ for some $\alpha > 1$, without the prior knowledge of $\alpha$ or an upper bound for $\mathbb{E}_t[|\ell_t(i)|^\alpha]$.

## Acknowledgements

EE acknowledges the financial support from the FAIR (Future Artificial Intelligence Research) project, funded by the NextGenerationEU program within the PNRR-PE-AI scheme (M4C2, investment 1.3, line on Artificial Intelligence), the EU Horizon CL4-2022-HUMAN-02 research and innovation action under grant agreement 101120237, project ELIAS (European Lighthouse of AI for Sustainability), and the One Health Action Hub, University Task Force for the resilience of territorial ecosystems, funded by Università degli Studi di Milano (PSR 2021-GSA-Linea 6). AM has received funding from the European Research Council (ERC), under the European Union's Horizon 2020 research and innovation programme (Grant agreement No. 950180). This work was partially done while DvdH was at the University of Amsterdam supported by Netherlands Organization for Scientific Research (NWO), grant number VI.Vidi.192.095.

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

# Appendix Contents

## A   Technical Results for Section 4

In Section 4, we discussed lower bounds for the lower-order terms. These lower bounds are achieved by adopting specific binary random variables in the construction of the losses. This is shown in Lemma 4.1 and its proof is provided in what follows.

### A.1   Proof of Lemma 4.1

*Proof of Lemma 4.1.* The fact that $\mathbb{E}[X_j^2] = 1$ follows by an immediate calculation after observing that each $X_j$ is a binary random variable. By independence, we also have that $\mathbb{E}[\max_{j \in [n]} X_j] = \sqrt{n}\left(1 - (1 - \frac{1}{n})^n\right)$. The result now follows from observing that $1/2 < 1 - (1 - \frac{1}{n})^n \le 1$, where we used the inequality $1 - x \le \exp(-x)$. $\qquad\square$

### A.2   Remarks on Fréchet Distributions

Here, we now discuss another family of random variables that guarantee similar properties. This family is that of Fréchet distributions [Fréchet, 1927]—see also Muraleedharan et al. [2011] for reference.

We denote by $\mathrm{Fréchet}(\alpha, s, m)$ the Fréchet distribution whose parameters are the shape parameter $\alpha > 0$, the scale parameter $s > 0$, and the location (of the minimum) parameter $m \in \mathbb{R}$. Its cumulative distribution function (CDF) is

$$\mathbb{P}(X \le x) = \exp\left(-\left(\frac{x - m}{s}\right)^{-\alpha}\right)$$

for $x > m$, where $X \sim \mathrm{Fréchet}(\alpha, s, m)$. If $\alpha > 1$, its expected value corresponds to $\mathbb{E}[X] = m + s\Gamma(1 - 1/\alpha)$, where $\Gamma(z) = \int_0^\infty t^{z-1} \exp(-t)\mathrm{d}t$ for $z > 0$ is the Gamma function.

**Lemma A.1.** *Fix any $n \in \mathbb{N} \setminus \{0\}$. Let $X_1, \ldots, X_n \sim \mathrm{Fréchet}(\alpha, s, 0)$ be i.i.d. random variables with $\alpha > 1$ and $s > 0$. Then, for any $1 \le \beta < \alpha$ and any $j \in [n]$,*

$$\mathbb{E}[X_j^\beta] = s^\beta \Gamma\left(1 - \frac{\beta}{\alpha}\right).$$

*Furthermore,*

$$\mathbb{E}[\max\{X_1, \ldots, X_n\}] = n^{1/\alpha} s \Gamma\left(1 - \frac{1}{\alpha}\right) .$$

*Proof.* All results are standard. Here we provide explicit calculations for completeness. First, observe that for any $X \sim \text{Fréchet}(\alpha, s, 0)$ we have that

$$\mathbb{P}\left(X^\beta \leq x\right) = \mathbb{P}\left(X \leq x^{1/\beta}\right) = \exp\left(-\left(\frac{x^{1/\beta}}{s}\right)^{-\alpha}\right) = \exp\left(-\left(\frac{x}{s^\beta}\right)^{-\alpha/\beta}\right) ,$$

showing that $X^\beta \sim \text{Fréchet}(\alpha/\beta, s^\beta, 0)$. As a consequence, $\mathbb{E}[X^\beta] = s^\beta \Gamma(1 - \beta/\alpha)$ since $\alpha/\beta > 1$.

Second, let $Y = \max\{X_1, \ldots, X_n\}$ and $y > 0$. By independence, one has

$$\mathbb{P}\left(Y \leq y\right) = \mathbb{P}\left(\bigcap_{j \in [n]} \{X_j \leq y\}\right) = \prod_{j \in [n]} \mathbb{P}\left(X_j \leq y\right)$$
$$= \prod_{j \in [n]} \exp\left(-(y/s)^{-\alpha}\right) = \exp\left(-\left(\frac{y}{n^{1/\alpha}s}\right)^{-\alpha}\right) ,$$

which means that $Y \sim \text{Fréchet}(\alpha, n^{1/\alpha}s, 0)$. Since $\alpha > 1$, its expectation is thus $\mathbb{E}[Y] = n^{1/\alpha} s \Gamma\left(1 - 1/\alpha\right)$. $\qquad\square$

Keeping the results from the lemma above in mind, we can consider a more specific case where the Fréchet random variables have scale $s = 1$ and $\alpha > 2$. Then, for $n = KT$ and $\beta = 2$, we clearly have that their second moment equals $\Gamma(1 - 2/\alpha)$, whereas the expectation of their maximum corresponds to $(KT)^{1/\alpha}\Gamma(1 - 1/\alpha)$. This shows that the i.i.d. Fréchet random variables as described above behave somewhat similarly to the random variables of Lemma 4.1 we employed in our lower bounds.

Moving back to the original construction from Lemma 4.1, we can adopt it to prove the main lower bound on any regret guarantee containing the $M_T$ term (or a constant fraction of it). This result from Section 4 is stated in Proposition 4.3 and here we provide its proof.

### A.3 Proof of Proposition 4.3

*Proof.* Let $\ell_t(i) = \mu_{t,i} + \xi_{t,i}\varepsilon_{t,i}$, where $\mu_{t,i} \in [-1, 1]$, $\xi_{t,i}$ is sampled from a Rademacher distribution, and $\varepsilon_{t,i}$ is sampled from the distribution specified in Lemma 4.1 with $n = KT$. Let $\sum_{t=1}^{T} \mu_{t,K} = 0$ and let $\sum_{t=1}^{T} \mu_{t,i} > 0$ for $i \neq K$. By assumption, we have that

$$R_T = \mathbb{E}\left[\sum_{t=1}^{T}\left(\sum_{i=1}^{K} p_t(i)\ell_t(i) - \ell_t(K)\right)\right] = \mathbb{E}\left[\sum_{t=1}^{T}\sum_{i=1}^{K} p_t(i)\ell_t(i)\right] \leq B_T^\star + \mathbb{E}\left[\max_{t \in [T], i \in [K]} |r_t(i)|\right] .$$

Observe that

$$
\mathbb{E}\left[\max_{t\in[T],i\in[K]}|r_t(i)|\right] = \mathbb{E}\left[\max_{t\in[T],i\in[K]}|\ell_t(i)-\langle p_t,\ell_t\rangle|\right]
$$

$$
\geq \mathbb{E}\left[\max_{t\in[T],i\in[K]}\Big(|\ell_t(i)-\ell_t(K)| - |\ell_t(K)-\langle p_t,\ell_t\rangle|\Big)\right]
$$

(reverse triangle inequality)

$$
= \mathbb{E}\left[\max_{t\in[T],i\in[K]}\Big(|\ell_t(i)-\ell_t(K)| - |r_t(K)|\Big)\right]
$$

$$
\geq \mathbb{E}\left[\max_{t\in[T],i\in[K]}|\ell_t(i)-\ell_t(K)|\right] - \mathbb{E}\left[\max_{t\in[T],i\in[K]}|r_t(i)|\right]
$$

$$
\geq \mathbb{E}\left[\max_{t\in[T],i\in[K]}\Big(|\ell_t(i)| - |\ell_t(K)|\Big)\right] - \mathbb{E}\left[\max_{t\in[T],i\in[K]}|r_t(i)|\right]
$$

(reverse triangle inequality)

$$
\geq \mathbb{E}\left[\max_{t\in[T],i\in[K]}|\ell_t(i)|\right] - \mathbb{E}\left[\max_{t\in[T]}|\ell_t(K)|\right] - \mathbb{E}\left[\max_{t\in[T],i\in[K]}|r_t(i)|\right]
$$

$$
\geq \mathbb{E}\left[\max_{t\in[T],i\in[K]}|\varepsilon_{t,i}|\right] - \mathbb{E}\left[\max_{t\in[T]}|\varepsilon_{t,K}|\right] - \mathbb{E}\left[\max_{t\in[T],i\in[K]}|r_t(i)|\right] - 2 .
$$

By carefully using the properties of the random variables $\varepsilon_{t,i}$ from Lemma 4.1, we have that

$$
\mathbb{E}\left[\max_{t\in[T],i\in[K]}|\varepsilon_{t,i}|\right] \geq \left(1-\frac{1}{e}\right)\sqrt{KT}
$$

as well as

$$
\mathbb{E}\left[\max_{t\in[T]}|\varepsilon_{t,j}|\right] = \sqrt{KT}\left(1-\left(1-\frac{1}{KT}\right)^T\right) \leq \sqrt{\frac{T}{K}}
$$

for any fixed $j\in[K]$, where the last inequality follows from the fact that $(1-x)^n \geq 1-nx$ for all $x\leq 1$ and all $n\geq 1$. Therefore, by rearranging the terms in the previous inequality we obtain

$$
\mathbb{E}\left[\max_{t\in[T],i\in[K]}|r_t(i)|\right] \geq \frac{\sqrt{T}}{2}\left(\left(1-\frac{1}{e}\right)\sqrt{K}-\frac{1}{\sqrt{K}}\right) - 1 \geq \frac{\sqrt{KT}}{16} - 1 ,
$$

where the last inequality holds for any $K\geq 2$. Hence, we immediately have that

$$
B_T^\star + \mathbb{E}\left[\max_{t\in[T],i\in[K]}|r_t(i)|\right] \geq B_T^\star + \frac{1}{16}\sqrt{KT} - 1 .
$$

Finally, we have

$$
\mathbb{E}_t[\ell_t(i)^2] = \mu_{t,i}^2 + 2\mu_{t,i}\,\mathbb{E}_t[\varepsilon_{t,i}] + \mathbb{E}_t[\varepsilon_{t,i}^2] \leq 2 + 2\,\mathbb{E}_t[\varepsilon_{t,i}] = 2 + \frac{2}{\sqrt{KT}} \leq 3
$$

since $K,T\geq 2$, which concludes the proof. $\qquad\square$

### A.4 Discussion on analyses of existing algorithms

Our results from Section 4 show that the lower-order terms are a problematic component of the regret bounds of existing algorithms. In particular, we have shown that the lower-order terms can be larger than the ideal $\sqrt{\theta T\log K}$ in the case of heavy-tailed losses with bounded second moments. It is important to note that our general argument does not depend on the specific algorithm we consider, but rather on regret bounds involving terms such as $L_T$ or $M_T$. Hence, generally speaking, one may believe that the presence of $L_T$ or $M_T$ in the regret guarantees could be an artefact of a loose analysis. However, here we show that this is not the case for most existing algorithms.

To illustrate, let us consider a simplified view that captures the essential mechanism. The algorithms in Table 1 are effectively sophisticated modifications of the exponential weights (EW) algorithm designed to adapt to $H_T = \max_{t,i}|\ell_t(i)|$. Their sampling distributions are of the form $p_t(i) \propto$

$\exp(-\eta_t \ell_t(i))$, where we assume a fixed learning $\eta > 0$ for simplicity. A crucial aspect of these adaptive algorithms is choosing a learning rate related to $H_T$, often resembling $\eta = \min\{\alpha, \frac{1}{H_T}\}$, where $\alpha > 0$ represents some cap and where we assume that $H_T$ is known. Their choice ensures the control of terms like $\log(\mathbb{E}_{i \sim p_t}[\exp(-\eta \ell_t(i))])$, which is an essential component in any regret analysis of EW-based algorithms.

Now, we construct a sequence of losses for which the regret of such an algorithm is always at least $\mathbb{E}[\log(4/3)/\eta]$. Suppose $K \geq 5$ and let $\ell_t(i) = 1 + \varepsilon_{t,i}$ for each $i \in [K-1]$, where $\varepsilon_{t,i}$ follows the same distribution as in Lemma 4.1; for action $K$, we instead choose $\ell_t(K) = 0$. By Jensen's inequality and the definition of $p_t$, we have that

$$
\begin{aligned}
R_T &= \mathbb{E}\left[\sum_{t=1}^{T} \langle p_t, \ell_T \rangle\right] \\
&\geq \mathbb{E}\left[-\frac{1}{\eta}\sum_{t=1}^{T} \log\big(\mathbb{E}_{i \sim p_t}\left[\exp(-\eta \ell_t(i))\right]\big)\right] \\
&= \mathbb{E}\left[\frac{1}{\eta}\log(K) - \frac{1}{\eta}\log\left(\sum_{i=1}^{K}\exp\left(-\eta\sum_{t=1}^{T}\ell_t(i)\right)\right)\right] \\
&\geq \mathbb{E}\left[\frac{1}{\eta}\log(K) - \frac{1}{\eta}\log\big(1 + (K-1)\exp(-\eta T)\big)\right],
\end{aligned}
$$

where the last inequality is due to the definition of $\ell_t(i)$. It is safe to assume that $\eta T < 1$, as otherwise we would obtain a vacuous upper bound from the standard EW analysis. Therefore, we have that

$$
\begin{aligned}
R_T &\geq \mathbb{E}\left[\frac{1}{\eta}\log(K) - \frac{1}{\eta}\log(1 + (K-1)/2)\right] \\
&\geq \mathbb{E}\left[\frac{1}{\eta}\log(K) - \frac{1}{\eta}\log(3(K-1)/4)\right] \\
&\geq \mathbb{E}\left[\frac{1}{\eta}\right]\log(4/3) \\
&\geq L_T \log(4/3),
\end{aligned}
$$

where the last inequality follows from the definitions of $\eta$ and $L_T = \mathbb{E}[H_T]$. This suggests the regret is lower bounded by a term proportional to the maximum loss range. For algorithms like Squint that run EW on the surrogate losses like $\ell_t(i) - \eta \ell_t(i)^2$, a similar but slightly more involved argument can be made. Consequently, the $L_T$ (or $M_T$) term appears to be unavoidable in the regret of these algorithms.

Similarly, regarding Section 4.2, the existing algorithms for the squared loss setting are also fundamentally related to EW, adapted to quadratic losses. Their learning rates are typically dependent on the range $\max_t |\bar{\mathbf{z}}_t - y_t|$ in a manner analogous to the dependence on $H_T$ described above. Therefore, we believe a similar limitation applies, making it unlikely that a different analysis of previous algorithms could circumvent the dependence on lower-order terms in their regret.

## B    Regret Analysis for Online Mirror Descent

In Algorithm 1 we only update $p_t$ if $\sum_{s \leq t}\bar{v}_t > 0$. The only case where this is not true is if in all rounds up to and including round $t$, $r_t(i) = 0$ for all $i \in [K]$, in which case we can simply ignore these rounds in the analysis, which is what we do: throughout this section we assume that $\bar{v}_1 > 0$ without loss of generality.

### B.1    Adversarial Environments

We now analyze the regret incurred by Algorithm 1 for adversarial and self-bounded environments. We start by reminding a standard lemma.

**Lemma B.1** (3-point identity, Chen and Teboulle, 1993). *Let $X \subseteq \mathbb{R}^d$ be a non-empty convex set, $\psi : X \to \mathbb{R}$ be strictly convex and differentiable on* $\mathrm{int}(X) \neq \emptyset$, *and $B_\psi$ be the associated Bregman divergence. Then for any $x \in X$ and any $y, z \in \mathrm{int}(X)$, the following holds*

$$B_\psi(x, y) = B_\psi(x, z) + B_\psi(z, y) + \langle x - z, \nabla\psi(z) - \nabla\psi(y) \rangle .$$

We then have the following result about the regret of Algorithm 1.

**Theorem B.2.** *Consider Algorithm 1 run with parameters $\alpha \in (0, 1]$ and $\beta > 0$. Then, with probability one, for any $i^\star \in [K]$ we have*

$$\sum_{t=1}^{T} \langle p_t - e_{i^\star}, \ell_t \rangle$$

$$\leq \left( \sqrt{\alpha T} + 5\beta + \frac{4 + \log(K/\alpha)}{\beta} \right) \sqrt{\sum_{t=1}^{T} \bar{v}_t} + \left( \sqrt{\alpha T} + \frac{\log(K/\alpha)}{\beta} + 2\beta \right) \sqrt{\sum_{t=1}^{T} v_t(i^\star)} .$$

*Furthermore, setting $\alpha = \frac{1}{T}$ and $\beta = \sqrt{\log(KT)}$ gives*

$$\sum_{t=1}^{T} \langle p_t - e_{i^\star}, \ell_t \rangle = \mathcal{O} \left( \sqrt{\log(KT) \left( \sum_{t=1}^{T} \bar{v}_t + \sum_{t=1}^{T} v_t(i^\star) \right)} \right) .$$

*Proof.* Let $i^\star \in [K]$ be our point-mass comparator, let $\bar{\ell}_t = -r_t$ for any $t \in [T]$, and recall that $\mathcal{P}_\alpha = \{ p \in \mathcal{P} : p \geq \frac{\alpha}{K}\mathbb{1} \}$ is the truncated probability simplex, for any $\alpha > 0$. Define $\mathrm{trunc}_T(x) = \sum_{t=1}^{T} \langle x, \bar{\ell}_t - \widetilde{\ell}_t \rangle$ to denote the cost of truncating the losses for any $x \in \mathbb{R}^K$ and, by convention, let $\mathrm{trunc}_T(p_{1:T}) = \sum_{t=1}^{T} \langle p_t, \bar{\ell}_t - \widetilde{\ell}_t \rangle$ where $p_{1:T} = (p_1, \ldots, p_T)$. Moreover, define $\widetilde{R}_T(q) = \sum_{t=1}^{T} \langle p_t - q, \widetilde{\ell}_t \rangle$ for any $q \in \mathcal{P}$. We consider the following regret decomposition:

$$\sum_{t=1}^{T} \langle p_t - e_{i^\star}, \ell_t \rangle = \sum_{t=1}^{T} \left\langle p_t - e_{i^\star}, \ell_t - \widetilde{\ell}_t \right\rangle + \sum_{t=1}^{T} \left\langle p_t - e_{i^\star}, \widetilde{\ell}_t \right\rangle$$

$$= \underbrace{\sum_{t=1}^{T} \left\langle p_t - e_{i^\star}, \bar{\ell}_t - \widetilde{\ell}_t \right\rangle}_{= \mathrm{trunc}_T(p_{1:T}) + \mathrm{trunc}_T(-e_{i^\star})} + \underbrace{\sum_{t=1}^{T} \left\langle p_t - e_{i^\star}, \widetilde{\ell}_t \right\rangle}_{= \widetilde{R}_T(e_{i^\star})},$$

where we added a constant vector $\langle p_t, \ell_t \rangle \mathbb{1} = \ell_t - \bar{\ell}_t$ to $\ell_t$ at each term of the first sum in the second equality because, for any $t$, both $p_t$ and $e_{i^\star}$ are probability distributions and thus $\langle p_t - e_{i^\star}, c \cdot \mathbb{1} \rangle = c - c = 0$ for any $c \in \mathbb{R}$. The first term is just the cost of truncating the losses, and the second is just the regret of OMD on the truncated losses. We start with the latter.

**Step 1: control the regret of OMD.** Fix any $t \in [T]$. Note that by the first-order optimality condition of $p_{t+1}$ we have, for any $u \in \mathcal{P}_\alpha$,

$$\left\langle \widetilde{\ell}_t + \nabla\psi_t(p_{t+1}) - \nabla\psi_t(p_t), u - p_{t+1} \right\rangle \geq 0.$$

Following a simple sequence of derivations also referred to as the 3-point identity (see Lemma B.1), the inequality above can be rewritten as

$$\left\langle \widetilde{\ell}_t, u - p_{t+1} \right\rangle - (D_t(u\|p_{t+1}) + D_t(p_{t+1}\|p_t) - D_t(u\|p_t)) \geq 0. \tag{6}$$

This inequality cannot be applied to $\widetilde{R}_T(e_{i^\star})$ since $e_{i^\star}$ does not belong to $\mathcal{P}_\alpha$. Instead, we use a mixture of $e_{i^\star}$ and the uniform distribution, $u_{i^\star} = (1 - \alpha) e_{i^\star} + \frac{\alpha}{K}\mathbb{1} \in \mathcal{P}_\alpha$, and notice that by

definition of $u_{i^\star}$ and a triangle inequality

$$
\begin{aligned}
\widetilde{R}_T\left(e_{i^\star}\right) - \widetilde{R}_T\left(u_{i^\star}\right) &= \sum_{t=1}^{T}\left\langle u_{i^\star} - e_{i^\star}, \widetilde{\ell}_t\right\rangle \\
&= \alpha\left\langle \frac{1}{K}\mathbb{1} - e_{i^\star}, \sum_{t=1}^{T}\widetilde{\ell}_t\right\rangle && \text{(definition of } u_{i^\star}\text{)} \\
&\leq \alpha\sum_{i=1}^{K}\frac{1}{K}\sum_{t=1}^{T}\left|\widetilde{\ell}_t\left(i\right)\right| + \alpha\sum_{t=1}^{T}\left|\widetilde{\ell}_t\left(i^\star\right)\right| .
\end{aligned}
$$

Next, following up with Cauchy-Schwarz inequality,

$$
\begin{aligned}
\widetilde{R}_T\left(e_{i^\star}\right) - \widetilde{R}_T\left(u_{i^\star}\right) &\leq \alpha\sum_{i=1}^{K}\frac{1}{K}\sqrt{T\sum_{t=1}^{T}\widetilde{\ell}_t\left(i\right)^2} + \alpha\sqrt{T\sum_{t=1}^{T}\widetilde{\ell}_t\left(i^\star\right)^2} \\
&\leq \alpha\sqrt{T\sum_{t=1}^{T}\sum_{i=1}^{K}\frac{1}{K}\widetilde{\ell}_t\left(i\right)^2} + \alpha\sqrt{T\sum_{t=1}^{T}\widetilde{\ell}_t\left(i^\star\right)^2} && \text{(Jensen's inequality)} \\
&\leq \sqrt{\alpha T\sum_{t=1}^{T}\sum_{i=1}^{K}p_t\left(i\right)\widetilde{\ell}_t\left(i\right)^2} + \alpha\sqrt{T\sum_{t=1}^{T}\widetilde{\ell}_t\left(i^\star\right)^2} && (p_t \in \mathcal{P}_\alpha) \\
&\leq \sqrt{\alpha T}\left(\sqrt{\sum_{t=1}^{T}\bar{v}_t} + \sqrt{\sum_{t=1}^{T}v_t\left(i^\star\right)}\right) . && \text{(Cauchy-Schwarz, } \widetilde{\ell}_t^2 \leq \bar{\ell}_t^2\text{)}
\end{aligned}
$$

Therefore, we can focus on bounding $\widetilde{R}_T\left(u_{i^\star}\right)$. From Equation (6), we have

$$
\widetilde{R}_T\left(u_{i^\star}\right) \leq \sum_{t=1}^{T}\left(\left\langle p_t - p_{t+1}, \widetilde{\ell}_t\right\rangle - D_t\left(p_{t+1}\|p_t\right) + D_t\left(u_{i^\star}\|p_t\right) - D_t\left(u_{i^\star}\|p_{t+1}\right)\right).
$$

The first difference within the sum can be analyzed with local norms arguments [Orabona, 2025, Section 6.5]. However, we do not have control over the range of the losses, thus we cannot use the standard arguments and instead resort to specific learning rates to account for this. On the other hand, the second difference is almost a telescoping sum and requires a more careful analysis for the same reason. We start with the first. Denote $\widetilde{p}_{t+1} = \arg\min_{p \in \mathbb{R}_{\geq 0}^K}\left\{\left\langle p, \widetilde{\ell}_t\right\rangle + D_t\left(p\|p_t\right)\right\}$ the minimizer of the unconstrained optimization problem, and note that for any $i \in [K]$, $\widetilde{p}_{t+1}\left(i\right) = p_t\left(i\right)\exp\left(-\eta_{t,i}\widetilde{\ell}_t\left(i\right)\right) \leq 3p_t\left(i\right)$, where the inequality holds by definition of the losses which satisfy $\left|\widetilde{\ell}_t\left(i\right)\right| \leq \frac{1}{\eta_{t,i}}$. Since the function $\psi_t$ is twice differentiable on $\mathbb{R}_{>0}^K$, by Taylor's theorem there exists $z_t \in [p_t, \widetilde{p}_{t+1}]$ such that $D_t\left(\widetilde{p}_{t+1}\|p_t\right) = \frac{1}{2}\left\|\widetilde{p}_{t+1} - p_t\right\|_{\nabla^2\psi_t(z_t)}^2$. Therefore, the first difference in the inequality above becomes, for any $t \in [T]$,

$$
\begin{aligned}
\left\langle p_t - p_{t+1}, \widetilde{\ell}_t\right\rangle - D_t\left(p_{t+1}\|p_t\right) &\leq \left\langle p_t - \widetilde{p}_{t+1}, \widetilde{\ell}_t\right\rangle - D_t\left(\widetilde{p}_{t+1}\|p_t\right) && \text{(optimality of } \widetilde{p}_{t+1}\text{)} \\
&= \left\langle p_t - \widetilde{p}_{t+1}, \widetilde{\ell}_t\right\rangle - \frac{1}{2}\left\|\widetilde{p}_{t+1} - p_t\right\|_{\nabla^2\psi_t(z_t)}^2 && \text{(Taylor's theorem)} \\
&\leq \max_{x \in \mathbb{R}^K}\left\{\left\langle x, \widetilde{\ell}_t\right\rangle - \frac{1}{2}\|x\|_{\nabla^2\psi_t(z_t)}\right\} \\
&= \frac{1}{2}\left\|\widetilde{\ell}_t\right\|_{(\nabla^2\psi_t(z_t))^{-1}}^2 ,
\end{aligned}
$$

where the last step follows from the Fenchel-Young inequality applied to the convex function $\frac{1}{2} \left\| \cdot \right\|_{\nabla^2 \psi_t(z_t)}$. Noting that $\nabla^2 \psi_t(z_t) = \mathrm{Diag}\left( \eta_t \odot z_t \right)^{-1}$, we further have

$$
\begin{aligned}
\left\langle p_t - p_{t+1}, \widetilde{\ell}_t \right\rangle - D_t\left( p_{t+1} \| p_t \right) &\leq \frac{1}{2} \sum_{i=1}^{K} \eta_{t,i} z_t\left( i \right) \widetilde{\ell}_t\left( i \right)^2 \\
&\leq \frac{3}{2} \sum_{i=1}^{K} \eta_{t,i} p_t\left( i \right) \widetilde{\ell}_t\left( i \right)^2 && \left( z_t \leq 3 p_t \right) \\
&= \frac{3\beta}{2} \sum_{i=1}^{K} \frac{p_t\left( i \right) \widetilde{\ell}_t\left( i \right)^2}{\sqrt{\max\left\{ \sum_{s=1}^{t} \bar{v}_s, \sum_{s=1}^{t} v_s\left( i \right) \right\}}} && \text{(definition of } \eta_t\text{)} \\
&\leq \frac{3\beta}{2} \cdot \frac{\bar{v}_t}{\sqrt{\sum_{s=1}^{t} \bar{v}_s}} && \left( \widetilde{\ell}_t^2 \leq \bar{\ell}_t^2 \right).
\end{aligned}
$$

Summing up for all $t \in [T]$, we obtain

$$
\sum_{t=1}^{T} \left\langle p_t - p_{t+1}, \widetilde{\ell}_t \right\rangle - D_t\left( p_{t+1} \| p_t \right) \leq \frac{3\beta}{2} \sum_{t=1}^{T} \frac{\bar{v}_t}{\sqrt{\sum_{s=1}^{t} \bar{v}_s}} \leq 3\beta \sqrt{\sum_{t=1}^{T} \bar{v}_t},
$$

where the inequality follows from, *e.g.*, Orabona [2025, Lemma 4.13]. Moving on to the second difference, we have

$$
\begin{aligned}
&\sum_{t=1}^{T} \left( D_t\left( u_{i^\star} \| p_t \right) - D_t\left( u_{i^\star} \| p_{t+1} \right) \right) \\
&= D_T\left( u_{i^\star} \| p_T \right) - D_T\left( u_{i^\star} \| p_{T+1} \right) + \sum_{t=1}^{T-1} \left( D_t\left( u_{i^\star} \| p_t \right) - D_t\left( u_{i^\star} \| p_{t+1} \right) \right) \\
&= D_1\left( u_{i^\star} \| p_1 \right) - D_T\left( u_{i^\star} \| p_{T+1} \right) + \sum_{t=2}^{T} \left( D_t\left( u_{i^\star} \| p_t \right) - D_{t-1}\left( u_{i^\star} \| p_t \right) \right) \\
&\leq D_1\left( u_{i^\star} \| p_1 \right) + \sum_{t=2}^{T} \left( D_t\left( u_{i^\star} \| p_t \right) - D_{t-1}\left( u_{i^\star} \| p_t \right) \right),
\end{aligned}
$$

where the second equality is due to a telescopic sum, and the last inequality is because $D_T\left( u_{i^\star} \| p_{T+1} \right) \geq 0$. The sum above is given by

$$
\begin{aligned}
&\sum_{t=2}^{T} \left( D_t\left( u_{i^\star} \| p_t \right) - D_{t-1}\left( u_{i^\star} \| p_t \right) \right) \\
&= \sum_{t=2}^{T} \sum_{i=1}^{K} \left( \frac{1}{\eta_{t,i}} - \frac{1}{\eta_{t-1,i}} \right) \left( u_{i^\star}\left( i \right) \log\left( \frac{u_{i^\star}\left( i \right)}{p_t\left( i \right)} \right) - u_{i^\star}\left( i \right) + p_t\left( i \right) \right) \\
&\leq \sum_{t=2}^{T} \sum_{i=1}^{K} \left( \frac{1}{\eta_{t,i}} - \frac{1}{\eta_{t-1,i}} \right) \left( u_{i^\star}\left( i \right) \log\left( \frac{u_{i^\star}\left( i \right)}{p_t\left( i \right)} \right) + p_t\left( i \right) \right),
\end{aligned}
$$

where the inequality is due to having $\eta_{t,i} \leq \eta_{t-1,i}$ and $u_{i^\star}(i) \geq 0$ for any $i$ and any $t$. Since $p_t \in \mathcal{P}_\alpha$, we have for any $i \neq i^\star$ that $\frac{u_{i^\star}(i)}{p_t(i)} \leq 1$, *i.e.*, $\log\left(\frac{u_{i^\star}(i)}{p_t(i)}\right) \leq 0$. Thus,

$$\sum_{t=2}^{T} \left(D_t\left(u_{i^\star} \| p_t\right) - D_{t-1}\left(u_{i^\star} \| p_t\right)\right)$$

$$\leq \sum_{t=2}^{T} \left(\frac{1}{\eta_{t,i^\star}} - \frac{1}{\eta_{t-1,i^\star}}\right) u_{i^\star}(i^\star) \log\left(\frac{u_{i^\star}(i^\star)}{p_t(i^\star)}\right) + \sum_{t=2}^{T} \sum_{i=1}^{K} \left(\frac{1}{\eta_{t,i}} - \frac{1}{\eta_{t-1,i}}\right) p_t(i)$$

$$\leq \frac{\log(K/\alpha)}{\eta_{T,i^\star}} + \sum_{t=1}^{T} \sum_{i=1}^{K} \left(\frac{1}{\eta_{t,i}} - \frac{1}{\eta_{t-1,i}}\right) p_t(i),$$

where we used $u_{i^\star}(i^\star) \leq 1$, $p_t(i^\star) \geq \frac{\alpha}{K}$, and $\eta_{1,i^\star} > 0$ for the first sum and added the non-negative term for $t = 1$ in the second sum. For the remaining sum, notice that for any $t, i$, we have $\frac{1}{\eta_{t,i}} - \frac{1}{\eta_{t-1,i}} = \eta_{t,i}\left(\frac{1}{\eta_{t,i}^2} - \frac{1}{\eta_{t,i}\,\eta_{t-1,i}}\right) \leq \eta_{t,i}\left(\frac{1}{\eta_{t,i}^2} - \frac{1}{\eta_{t-1,i}^2}\right)$ due to $\eta_{t,i} \leq \eta_{t-1,i}$. Using the definition of the learning rates, we get

$$\frac{1}{\eta_{t,i}} - \frac{1}{\eta_{t-1,i}} \leq \eta_{t,i}\left(\frac{1}{\eta_{t,i}^2} - \frac{1}{\eta_{t-1,i}^2}\right) \tag{7}$$

$$= \frac{\max\left\{\sum_{s \leq t} \bar{v}_s, \sum_{s \leq t} v_s(i)\right\} - \max\left\{\sum_{s < t} \bar{v}_s, \sum_{s < t} v_s(i)\right\}}{\beta \sqrt{\max\left\{\sum_{s \leq t} \bar{v}_s, \sum_{s \leq t} v_s(i)\right\}}}$$

$$\leq \frac{\max\{\bar{v}_t, v_t(i)\}}{\beta \sqrt{\max\left\{\sum_{s \leq t} \bar{v}_s, \sum_{s \leq t} v_s(i)\right\}}}$$

$$\leq \frac{\bar{v}_t + v_t(i)}{\beta \sqrt{\sum_{s \leq t} \bar{v}_s}} . \tag{8}$$

Therefore, adding everything up

$$\sum_{t=1}^{T} \sum_{i=1}^{K} \left(\frac{1}{\eta_{t,i}} - \frac{1}{\eta_{t-1,i}}\right) p_t(i) \leq \sum_{t=1}^{T} \sum_{i=1}^{K} \frac{\bar{v}_t + v_t(i)}{\beta \sqrt{\sum_{s \leq t} \bar{v}_s}} p_t(i)$$

$$= \frac{2}{\beta} \sum_{t=1}^{T} \frac{\bar{v}_t}{\sqrt{\sum_{s \leq t} \bar{v}_s}} \qquad (p_t \in \mathcal{P}, \text{ definition } \bar{v}_t)$$

$$\leq \frac{4}{\beta} \sqrt{\sum_{t=1}^{T} \bar{v}_t} ,$$

where the last inequality follows again from Orabona [2025, Lemma 4.13]. Putting it back into the previous inequality, we get

$$\sum_{t=2}^{T} \left(D_t\left(u_{i^\star} \| p_t\right) - D_{t-1}\left(u_{i^\star} \| p_t\right)\right) \leq \frac{\log(K/\alpha)}{\eta_{T,i^\star}} + \frac{4}{\beta} \sqrt{\sum_{t=1}^{T} \bar{v}_t}$$

$$\leq \frac{\log(K/\alpha)}{\beta} \left(\sqrt{\sum_{t=1}^{T} v_t(i^\star)} + \sqrt{\sum_{t=1}^{T} \bar{v}_t}\right) + \frac{4}{\beta} \sqrt{\sum_{t=1}^{T} \bar{v}_t} .$$

Finally, the regret incurred by OMD on the truncated losses is bounded by

$$\widetilde{R}_T\left(e_{i^\star}\right) \leq \left(\sqrt{\alpha T} + 3\beta + \frac{4 + \log(K/\alpha)}{\beta}\right) \sqrt{\sum_{t=1}^{T} \bar{v}_t} + \left(\sqrt{\alpha T} + \frac{\log(K/\alpha)}{\beta}\right) \sqrt{\sum_{t=1}^{T} v_t(i^\star)} .$$

**Step 2: control the cost of truncation.** We have

$$
\begin{aligned}
\texttt{trunc}_T\left(p_{1:T}\right) &= \sum_{t=1}^{T}\sum_{i=1}^{K} p_t\left(i\right)\left(\bar{\ell}_t\left(i\right) - \widetilde{\ell}_t\left(i\right)\right) \\
&= \sum_{t=1}^{T}\sum_{i=1}^{K} p_t\left(i\right)\bar{\ell}_t\left(i\right)\mathbb{1}\left(\left|\bar{\ell}_t\left(i\right)\right| > \eta_{t,i}^{-1}\right) &&\text{(definition of } \widetilde{\ell}_t) \\
&\leq \sum_{t=1}^{T}\sum_{i=1}^{K} p_t\left(i\right)\frac{\bar{\ell}_t\left(i\right)^2}{\left|\bar{\ell}_t\left(i\right)\right|}\mathbb{1}\left(\left|\bar{\ell}_t\left(i\right)\right| > \eta_{t,i}^{-1}\right) &&(p_t \geq 0) \\
&\leq \sum_{t=1}^{T}\sum_{i=1}^{K} p_t\left(i\right)\eta_{t,i}\bar{\ell}_t\left(i\right)^2.
\end{aligned}
$$

Plugging the definition of the learning rates $\eta_t$, we obtain

$$
\begin{aligned}
\texttt{trunc}_T\left(p_{1:T}\right) &\leq \beta\sum_{t=1}^{T}\sum_{i=1}^{K} p_t\left(i\right)\frac{\bar{\ell}_t\left(i\right)^2}{\sqrt{\max\left\{\sum_{s=1}^{t}\bar{v}_s, \sum_{s=1}^{t} v_s\left(i\right)\right\}}} &&\text{(definition of } \eta_t) \\
&\leq \beta\sum_{t=1}^{T}\sum_{i=1}^{K}\frac{p_t\left(i\right)\bar{\ell}_t\left(i\right)^2}{\sqrt{\sum_{s=1}^{t}\bar{v}_s}} \\
&= \beta\sum_{t=1}^{T}\frac{\bar{v}_t}{\sqrt{\sum_{s=1}^{t}\bar{v}_s}} \\
&\leq 2\beta\sqrt{\sum_{t=1}^{T}\bar{v}_t},
\end{aligned}
$$

where the last inequality follows from Orabona [2025, Lemma 4.13]. Likewise,

$$
\texttt{trunc}_T\left(-e_{i^\star}\right) \leq 2\beta\sqrt{\sum_{t=1}^{T} v_t\left(i^\star\right)}.
$$

Overall, our regret is bounded by

$$
\sum_{t=1}^{T}\langle p_t - e_{i^\star}, \ell_t\rangle
$$

$$
\leq \left(\sqrt{\alpha T} + 5\beta + \frac{4 + \log\left(K/\alpha\right)}{\beta}\right)\sqrt{\sum_{t=1}^{T}\bar{v}_t} + \left(\sqrt{\alpha T} + \frac{\log\left(K/\alpha\right)}{\beta} + 2\beta\right)\sqrt{\sum_{t=1}^{T} v_t\left(i^\star\right)}. \quad\square
$$

Theorem B.2 is a more general result and it is indeed stronger than what we originally stated in Theorem 2.4. In particular, we are able to show that the former result implies the latter under Assumption 2.1. This is illustrated by the following corollary.

**Corollary B.3.** *Suppose Assumption 2.1 holds. Then, Algorithm 1 with $\alpha = \frac{1}{T}$ and $\beta = \sqrt{\log(KT)}$ guarantees*

$$
R_T = \mathcal{O}\left(\sqrt{\theta T \log(KT)}\right).
$$

*Proof.* Recall that, by Theorem B.2, Algorithm 1 with $\alpha = \frac{1}{T}$ and $\beta = \sqrt{\log(K)}$ already guarantees

$$
\sum_{t=1}^{T}\langle p_t - e_{i^\star}, \ell_t\rangle = \mathcal{O}\left(\sqrt{\log(KT)\left(\sum_{t=1}^{T}\bar{v}_t + \sum_{t=1}^{T} v_t(i^\star)\right)}\right) \tag{9}
$$

for any sequence of losses. Now, focus on the $\bar{v}_t$ and $v_t(i)$ terms. First, for any $i \in [K]$, we can observe that $v_t(i)$ satisfies

$$
\mathbb{E}_t[v_t(i)] = \mathbb{E}_t\left[\left(\ell_t(i) - \sum_{j=1}^{K} p_t(j)\ell_t(j)\right)^2\right] \qquad \text{(definition of } v_t(i))
$$

$$
= \mathbb{E}_t\left[\left(\sum_{j=1}^{K} p_t(j)\left(\ell_t(i) - \ell_t(j)\right)\right)^2\right]
$$

$$
\le \mathbb{E}_t\left[\sum_{j=1}^{K} p_t(j)(\ell_t(i) - \ell_t(j))^2\right] \qquad \text{(Jensen's inequality)}
$$

$$
\le 2\,\mathbb{E}_t[\ell_t(i)^2] + 2\,\mathbb{E}_t\left[\sum_{j=1}^{K} p_t(j)\ell_t(j)^2\right] \qquad \text{(using } (a-b)^2 \le 2a^2 + 2b^2 \text{ and } p_t \in \mathcal{P})
$$

$$
= 2\,\mathbb{E}_t[\ell_t(i)^2] + 2\sum_{j=1}^{K} p_t(j)\,\mathbb{E}_t\left[\ell_t(j)^2\right]
$$

$$
\le 4\theta\,,
$$

where the last inequality follows by Assumption 2.1 and the fact that $p_t \in \mathcal{P}$. Then, we can move to $\bar{v}_t$ and notice that, by its definition, it satisfies

$$
\mathbb{E}_t[\bar{v}_t] = \mathbb{E}_t\left[\sum_{i=1}^{K} p_t(i)v_t(i)\right]
$$

$$
= \mathbb{E}_t\left[\sum_{i=1}^{K} p_t(i)\left(\ell_t(i) - \sum_{j=1}^{K} p_t(j)\ell_t(j)\right)^2\right] \qquad \text{(definition of } v_t(i))
$$

$$
\le \mathbb{E}_t\left[\sum_{i=1}^{K} p_t(i)\ell_t(i)^2\right] \qquad \text{(using } \mathbb{V}[X] \le \mathbb{E}[X^2])
$$

$$
= \sum_{i=1}^{K} p_t(i)\,\mathbb{E}_t\left[\ell_t(i)^2\right]
$$

$$
\le \theta\,,
$$

where the last inequality holds by both Assumption 2.1 and the fact that $p_t \in \mathcal{P}$.

At this point, we can observe that $R_T = \max_{i^\star \in [K]} \mathbb{E}[\langle p_t - e_{i^\star}, \ell_t\rangle]$, i.e., it corresponds to the expected value of the left-hand side of Equation (9). We can then focus on the expected value of its right-hand side (ignoring constant factors) and, by applying Jensen's inequality with respect to the square root and the tower rule of expectation, we infer that

$$
\mathbb{E}\left[\sqrt{\log(KT)\sum_{t=1}^{T}\left(\bar{v}_t + v_t(i^\star)\right)}\right] \le \sqrt{\log(KT)\,\mathbb{E}\left[\sum_{t=1}^{T}\left(\mathbb{E}_t\left[\bar{v}_t\right] + \mathbb{E}_t\left[v_t(i^\star)\right]\right)\right]}
$$

$$
\le \sqrt{5\theta T \log(KT)}\,.
$$

This concludes the proof. $\qquad\square$

## B.2 Self-Bounded Environments

In this section, we provide a regret bound for self-bounded environments defined in Zimmert and Seldin [2021].

**Theorem B.4.** *Let Assumption 2.1, 2.3 hold, and consider Algorithm 1 run with parameters $\alpha \in (0, 1]$, $\beta > 0$. We denote $C_3(K, T) = 2\sqrt{\alpha T} + 7\beta + \frac{4 + 2\log(K/\alpha)}{\beta}$. Then, we have*

$$\mathbb{E}[R_T] \leq \begin{cases} \frac{16\, C_3(K,T)^2 \theta}{\Delta_{\min}} + \frac{8\, C_3(K,T)\sqrt{\theta C}}{\sqrt{\Delta_{\min}}} & \text{if } C \leq \frac{16\, C_3(K,T)^2 \theta}{\Delta_{\min}} \\ \frac{8\, C_3(K,T)\sqrt{\theta C}}{\sqrt{\Delta_{\min}}} & \text{otherwise} \end{cases}.$$

*Furthermore, setting $\alpha = \frac{1}{T}$ and $\beta = \sqrt{\log(KT)}$ gives $C_3(K, T) = \mathcal{O}\left(\sqrt{\log(KT)}\right)$ and*

$$\mathbb{E}[R_T] \leq \begin{cases} \frac{3600\log(KT)\theta}{\Delta_{\min}} + 120\sqrt{\frac{\log(KT)\theta C}{\Delta_{\min}}} & \text{if } C \leq \frac{16\, C_3(K,T)^2 \theta}{\Delta_{\min}} \\ 120\sqrt{\frac{\log(KT)\theta C}{\Delta_{\min}}} & \text{otherwise} \end{cases}.$$

*Proof.* From Jensen's inequality applied to the regret bound proven for adversarial environments in Theorem B.2, we have

$$\mathbb{E}[R_T(e_{i^\star})] \leq C_1(K, T)\sqrt{\sum_{t=1}^{T}\sum_{i=1}^{K}\mathbb{E}\left[p_t(i)\left(\ell_t(i) - \sum_{j=1}^{K}p_t(j)\ell_t(j)\right)^2\right]}$$

$$+ C_2(K, T)\sqrt{\sum_{t=1}^{T}\mathbb{E}\left[\left(\ell_t(i^\star) - \sum_{j=1}^{K}p_t(j)\ell_t(j)\right)^2\right]},$$

where $C_1(K, T) = \sqrt{\alpha T} + 5\beta + \frac{4 + \log(K/\alpha)}{\beta}$, and $C_2(K, T) = \sqrt{\alpha T} + \frac{\log(K/\alpha)}{\beta} + 2\beta$. Observe that for any $t \in [T]$, the expectation in the first term is a variance that can be bounded by the second moment ($\mathbb{V}[X] \leq \mathbb{E}[X^2]$ for any random variable $X$)

$$\mathbb{E}_t\left[\sum_{i=1}^{K}p_t(i)\left(\ell_t(i) - \sum_{j=1}^{K}p_t(j)\ell_t(j)\right)^2\right]$$

$$= \mathbb{E}_t\left[\sum_{i=1}^{K}p_t(i)\left((\ell_t(i) - \ell_t(i^\star)) - \sum_{j=1}^{K}p_t(j)(\ell_t(j) - \ell_t(i^\star))\right)^2\right]$$

$$\leq \mathbb{E}_t\left[\sum_{i \neq i^\star}p_t(i)(\ell_t(i) - \ell_t(i^\star))^2\right].$$

Further, using the inequality $(a + b)^2 \leq 2(a^2 + b^2)$ for any $a, b \in \mathbb{R}$,

$$\mathbb{E}_t\left[\sum_{i=1}^{K}p_t(i)\left(\ell_t(i) - \sum_{j=1}^{K}p_t(j)\ell_t(j)\right)^2\right] \leq 2\mathbb{E}_t\left[\sum_{i \neq i^\star}p_t(i)\left(\ell_t(i)^2 + \ell_t(i^\star)^2\right)\right]$$

$$\leq 4\theta(1 - p_t(i^\star)),$$

where the last inequality is by Assumption 2.1 and $\langle p_t, \mathbb{1}\rangle = 1$. Likewise, for the second term we have by Jensen's inequality

$$\mathbb{E}_t\left[\left(\ell_t(i^\star) - \sum_{j=1}^{K}p_t(j)\ell_t(j)\right)^2\right] \leq \mathbb{E}_t\left[\sum_{j \neq i^\star}p_t(j)(\ell_t(i^\star) - \ell_t(j))^2\right]$$

$$\leq 2\mathbb{E}_t\left[\sum_{j \neq i^\star}p_t(j)\left(\ell_t(i^\star)^2 + \ell_t(j)^2\right)\right]$$

$$\leq 4\theta(1 - p_t(i^\star)),$$

where the last inequality is again by Assumption 2.1 and $\langle p_t, \mathbb{1} \rangle = 1$. Denote $\Delta_{\min} = \min_{i \neq i^\star} \Delta_i$, and $C_3(K,T) = C_1(K,T) + C_2(K,T)$. Combining the above and Assumption 2.3, we find that for any $\lambda \in (0,1]$,

$$\mathbb{E}[R_T] = (1+\lambda)\mathbb{E}[R_T] - \lambda\mathbb{E}[R_T]$$

$$\leq (1+\lambda)C_3(K,T)\sqrt{4\theta\,\mathbb{E}\left[\sum_{t=1}^{T}(1-p_t(i^\star))\right]} - \lambda\Delta_{\min}\mathbb{E}\left[\sum_{t=1}^{T}(1-p_t(i^\star))\right] + \lambda C$$

$$= \sqrt{4(1+\lambda)^2 C_3(K,T)^2\theta\,\mathbb{E}\left[\sum_{t=1}^{T}(1-p_t(i^\star))\right]} - \lambda\Delta_{\min}\mathbb{E}\left[\sum_{t=1}^{T}(1-p_t(i^\star))\right] + \lambda C.$$

Using the inequality of arithmetic and geometric means, *i.e.* $|ab| = \inf_{\gamma>0} \frac{a^2}{2\gamma} + \frac{\gamma b^2}{2}$,

$$\mathbb{E}[R_T] = \inf_{\gamma>0}\left\{\frac{4(1+\lambda)^2 C_3(K,T)^2\theta}{\gamma} + \gamma\mathbb{E}\left[\sum_{t=1}^{T}(1-p_t(i^\star))\right]\right\}$$

$$- \lambda\Delta_{\min}\mathbb{E}\left[\sum_{t=1}^{T}(1-p_t(i^\star))\right] + \lambda C$$

$$\leq \frac{4(1+\lambda)^2 C_3(K,T)^2\theta}{\lambda\Delta_{\min}} + \lambda C$$

$$\leq \frac{16\,C_3(K,T)^2\theta}{\lambda\Delta_{\min}} + \lambda C,$$

where the first inequality is by taking $\gamma = \lambda\Delta_{\min}$, and the second is by $1 + \lambda \leq 2$. In particular, setting $\lambda = \min\left\{1, \frac{4C_3(K,T)\sqrt{\theta}}{\sqrt{\Delta_{\min}C}}\right\}$ gives

$$\mathbb{E}[R_T] \leq \frac{16\,C_3(K,T)^2\theta}{\Delta_{\min}}\max\left\{1, \frac{\sqrt{\Delta_{\min}C}}{4C_3(K,T)\sqrt{\theta}}\right\} + \min\left\{1, \frac{4C_3(K,T)\sqrt{\theta}}{\sqrt{\Delta_{\min}C}}\right\}C$$

$$\leq \frac{16\,C_3(K,T)^2\theta}{\Delta_{\min}} + \frac{8\,C_3(K,T)\sqrt{\theta C}}{\sqrt{\Delta_{\min}}}.$$

If $C > \frac{16\,C_3(K,T)^2\theta}{\Delta_{\min}}$, then $\lambda = \frac{4C_3(K,T)\sqrt{\theta}}{\sqrt{\Delta_{\min}C}}$ and the previous inequality can be improved to

$$\mathbb{E}[R_T] \leq \frac{8\,C_3(K,T)\sqrt{\theta C}}{\sqrt{\Delta_{\min}}}.$$

Recall that $C_3(K,T) = 2\sqrt{\alpha T} + 7\beta + \frac{4+2\log(K/\alpha)}{\beta}$. Setting $\alpha = \frac{1}{T}$ and $\beta = \sqrt{\log(KT)}$ gives $C_3(K,T) \leq 6 + 9\sqrt{\log(KT)} \leq 15\sqrt{\log(KT)}$ since $\log(KT) \geq 1$. Finally, we obtain

$$\mathbb{E}[R_T] \leq \begin{cases} \frac{3600\log(KT)\theta}{\Delta_{\min}} + 120\sqrt{\frac{\log(KT)\theta C}{\Delta_{\min}}} & \text{if } C \leq \frac{16\,C_3(K,T)^2\theta}{\Delta_{\min}} \\ 120\sqrt{\frac{\log(KT)\theta C}{\Delta_{\min}}} & \text{otherwise} \end{cases}.$$

In the special case of a stochastic environment, $C = 0$ thus we set $\lambda = 1$ and obtain $\mathbb{E}[R_T] = \mathcal{O}\left(\frac{\theta\log(KT)}{\Delta_{\min}}\right)$. $\qquad\square$

## C  Regret Analysis for Follow The Regularized Leader

In Algorithm 2 we only update the prediction $p_t$ if $\sum_{s \leq t} v_s > 0$. The only case where this is not true is if in all rounds up to and including round $t$, $r_t(i) = 0$ for all $i \in [K]$, meaning that the cumulative regret up to round $t$ is null. In this case, we can simply ignore these rounds in the regret analysis, which is what we do: throughout this section we assume that $v_1 > 0$ without loss of generality.

**Theorem C.1.** *Consider Algorithm 2 with parameter $\beta > 0$, providing predictions $p_1, \ldots, p_T \in \mathcal{P}$ over a sequence of losses $\ell_1, \ldots, \ell_T$. Then, with probability one, for any $i^\star \in [K]$ we have*

$$\sum_{t=1}^{T} \langle p_t - e_{i^\star}, \ell_t \rangle$$

$$\leq \left( \frac{\log(K)}{\beta} + 2\beta \right) \sqrt{\sum_{t=1}^{T} v_t(i^\star)} + \left( \frac{5 + \log(K)}{\beta} + 5\beta \right) \sqrt{\sum_{t=1}^{T} \bar{v}_t} + \frac{1}{\beta} \sum_{i=1}^{K} \frac{1}{K} \sqrt{\sum_{t=1}^{T} v_t(i)} .$$

*Furthermore, setting $\beta = \sqrt{\log(K)}$ gives*

$$\sum_{t=1}^{T} \langle p_t - e_{i^\star}, \ell_t \rangle = \mathcal{O} \left( \sqrt{\log(K) \left( \sum_{t=1}^{T} \bar{v}_t + \sum_{t=1}^{T} v_t(i^\star) \right)} + \frac{1}{K} \sum_{i=1}^{K} \sqrt{\sum_{t=1}^{T} v_t(i)} \right) .$$

*Proof.* Let $i^\star \in [K]$ be our point-mass comparator and let $\bar{\ell}_t = -r_t$ for any $t \in [T]$. First, observe that the regret can be equivalently rewritten as

$$\sum_{t=1}^{T} \langle p_t - e_{i^\star}, \ell_t \rangle = \sum_{t=1}^{T} \langle p_t - e_{i^\star}, \bar{\ell}_t \rangle ;$$

this follows from the fact that replacing $\ell_t$ with $\bar{\ell}_t$ only leads to a difference $\langle p_t - e_{i^\star}, \ell_t - \bar{\ell}_t \rangle = \langle p_t - e_{i^\star}, c \cdot \mathbb{1} \rangle = \langle p_t, c \cdot \mathbb{1} \rangle - \langle e_{i^\star}, c \cdot \mathbb{1} \rangle = c - c = 0$ for the constant $c = \langle p_t, \ell_t \rangle$, by definition of $\bar{\ell}_t$. As in the proof of Theorem B.2, we take the same definitions of $\texttt{trunc}_T$ and $\widetilde{R}_T$, and consider the following regret decomposition:

$$\sum_{t=1}^{T} \langle p_t - e_{i^\star}, \ell_t \rangle = \underbrace{\sum_{t=1}^{T} \left\langle p_t - e_{i^\star}, \bar{\ell}_t - \widetilde{\ell}_t \right\rangle}_{= \texttt{trunc}_T(p_{1:T}) + \texttt{trunc}_T(-e_{i^\star})} + \underbrace{\sum_{t=1}^{T} \left\langle p_t - e_{i^\star}, \widetilde{\ell}_t \right\rangle}_{= \widetilde{R}_T(e_{i^\star})},$$

where the first term is the cost of truncating the losses, and the second is the regret of FTRL on the truncated losses $\widetilde{\ell}_1, \ldots, \widetilde{\ell}_T$. Let us first focus on the former term.

Before we proceed, one final remark is in order. Recall that Algorithm 2 performs predictions defined such that

$$p_t = \arg \min_{p \in \mathcal{P}} \sum_{s=1}^{t-1} \left\langle p, \widetilde{\ell}_s \right\rangle + D_{t-1}(p \| p_0)$$

for any $t > 1$, where $p_0 = \frac{1}{K} \mathbb{1} \in \mathcal{P}$ is the uniform distribution, while $p_1 = p_0$. Observe that we can define $\eta_{0,i} = \beta / \sqrt{\bar{v}_1}$ (never used nor set by Algorithm 2) for any $i \in [K]$ and, thus, we can equivalently denote $p_1$ as

$$p_1 \in \arg \min_{p \in \mathcal{P}} D_0(p \| p_0)$$

because $D_0(p \| p_0) = \frac{\sqrt{\bar{v}_1}}{\beta} \sum_{i=1}^{K} \big( p(i) \log(p(i)/p_0(i)) - p(i) + p_0(i) \big)$ is minimized at $p_0$. We remark that, while this step appears to require knowledge of $\bar{v}_1$ before computing the prediction $p_1$, it is only part of the analysis and not algorithmically performed.

**Step 1: control the cost of truncation.** We can begin by focusing on $\texttt{trunc}_T(p_{1:T})$. Observe that

$$\texttt{trunc}_T(p_{1:T}) = \sum_{t=1}^{T} \left\langle p_t, \bar{\ell}_t - \widetilde{\ell}_t \right\rangle$$

$$\leq \sum_{t=1}^{T} \sum_{i=1}^{K} p_t(i) \big| \bar{\ell}_t(i) - \widetilde{\ell}_t(i) \big|$$

$$= \sum_{t=1}^{T} \sum_{i=1}^{K} p_t(i) \left( |\bar{\ell}_t(i)| \left( 1 - \frac{b_{t-1}(i)}{b_t(i)} \right) \mathbb{1} \left( |\bar{\ell}_t(i)| \leq \frac{1}{\eta_{t,i}} \right) + |\bar{\ell}_t(i)| \, \mathbb{1} \left( |\bar{\ell}_t(i)| > \frac{1}{\eta_{t,i}} \right) \right),$$

where the last equality follows by definition of $\widetilde{\ell}_t(i)$, after observing that $b_{t-1}(i) \leq b_t(i)$. Now, by using the definitions of $b_t(i)$ and $\eta_{t,i}$, observe that

$$\sum_{i=1}^{K} p_t(i) \left|\bar{\ell}_t(i)\right| \left(1 - \frac{b_{t-1}(i)}{b_t(i)}\right) \mathbb{1}\left(\left|\bar{\ell}_t(i)\right| \leq \frac{1}{\eta_{t,i}}\right)$$

$$= \sum_{i=1}^{K} p_t(i) \frac{\left|\bar{\ell}_t(i)\right|}{b_t(i)} (b_t(i) - b_{t-1}(i)) \mathbb{1}\left(\left|\bar{\ell}_t(i)\right| \leq \frac{1}{\eta_{t,i}}\right)$$

$$= \sum_{i=1}^{K} p_t(i) \eta_{t,i} \left|\bar{\ell}_t(i)\right| \left(\frac{1}{\eta_{t,i}} - \frac{1}{\eta_{t-1,i}}\right) \mathbb{1}\left(\left|\bar{\ell}_t(i)\right| \leq \frac{1}{\eta_{t,i}}\right)$$

$$\leq \sum_{i=1}^{K} p_t(i) \left(\frac{1}{\eta_{t,i}} - \frac{1}{\eta_{t-1,i}}\right)$$

$$\leq \frac{2\bar{v}_t}{\beta \sqrt{\sum_{s \leq t} \bar{v}_s}} ,$$

for $t > 1$, where the last inequality holds by Equation (8) given that the learning rates $\eta_{t,i}$ have the same definition; the same bound holds similarly for $t = 1$ by observing that $b_0(i) = 0$ and, hence, we have

$$\sum_{i=1}^{K} p_1(i) |\bar{\ell}_1(i)| \mathbb{1}\left(|\bar{\ell}_1(i)| \leq \frac{1}{\eta_{1,i}}\right) \leq \frac{1}{\beta} \sum_{i=1}^{K} p_1(i) \sqrt{\max\{\bar{v}_1, v_1(i)\}}$$

$$\leq \frac{1}{\beta} \sqrt{\sum_{i=1}^{K} p_1(i) \max\{\bar{v}_1, v_1(i)\}} \leq \frac{\sqrt{2\bar{v}_1}}{\beta} \leq \frac{2\bar{v}_1}{\beta \sqrt{\bar{v}_1}} ,$$

where the second step follows by Jensen's inequality. At the same time, we have that

$$\sum_{i=1}^{K} p_t(i) \left|\bar{\ell}_t(i)\right| \mathbb{1}\left(\left|\bar{\ell}_t(i)\right| > \frac{1}{\eta_{t,i}}\right) = \sum_{i=1}^{K} p_t(i) \frac{\bar{\ell}_t(i)^2}{|\bar{\ell}_t(i)|} \mathbb{1}\left(\left|\bar{\ell}_t(i)\right| > \frac{1}{\eta_{t,i}}\right)$$

$$\leq \sum_{i=1}^{K} \eta_{t,i} p_t(i) \bar{\ell}_t(i)^2$$

$$\leq \frac{\beta \bar{v}_t}{\sqrt{\sum_{s \leq t} \bar{v}_s}} .$$

We can therefore combine the above inequalities and, together with Orabona [2025, Lemma 4.13], obtain that

$$\texttt{trunc}_T(p_{1:T}) \leq \left(\beta + \frac{2}{\beta}\right) \sum_{t=1}^{T} \frac{\bar{v}_t}{\sqrt{\sum_{s \leq t} \bar{v}_s}} \leq 2\left(\beta + \frac{2}{\beta}\right) \sqrt{\sum_{t=1}^{T} \bar{v}_t} .$$

Similarly, we can see that $\texttt{trunc}_T(-e_{i^\star})$ similarly satisfies

$$\texttt{trunc}_T(-e_{i^\star}) = \sum_{t=1}^{T} \left\langle -e_{i^\star}, \bar{\ell}_t - \widetilde{\ell}_t \right\rangle$$

$$\leq \sum_{t=1}^{T} \left|\bar{\ell}_t(i^\star) - \widetilde{\ell}_t(i^\star)\right|$$

$$= \sum_{t=1}^{T} \left(\left|\bar{\ell}_t(i^\star)\right| \left(1 - \frac{b_{t-1}(i^\star)}{b_t(i^\star)}\right) \mathbb{1}\left(\left|\bar{\ell}_t(i^\star)\right| \leq \frac{1}{\eta_{t,i^\star}}\right) + \left|\bar{\ell}_t(i^\star)\right| \mathbb{1}\left(\left|\bar{\ell}_t(i^\star)\right| > \frac{1}{\eta_{t,i^\star}}\right)\right).$$

Using similar calculations as before, it follows that

$$\sum_{t=1}^{T} |\bar{\ell}_t(i^\star)| \left(1 - \frac{b_{t-1}(i^\star)}{b_t(i^\star)}\right) \mathbb{1}\left(|\bar{\ell}_t(i^\star)| \le \frac{1}{\eta_{t,i^\star}}\right) \le \frac{1}{\beta} \sum_{t=1}^{T} (b_t(i^\star) - b_{t-1}(i^\star)) = \frac{1}{\eta_{T,i^\star}}$$

$$\le \frac{1}{\beta}\sqrt{\sum_{t=1}^{T} \bar{v}_t} + \frac{1}{\beta}\sqrt{\sum_{t=1}^{T} v_t(i^\star)},$$

where we used the subadditivity of the square root in the last inequality, and also that

$$\sum_{t=1}^{T} |\bar{\ell}_t(i^\star)| \, \mathbb{1}\left(|\bar{\ell}_t(i^\star)| > \frac{1}{\eta_{t,i^\star}}\right) \le \sum_{t=1}^{T} \eta_{t,i^\star} \bar{\ell}_t(i^\star)^2 \le \beta \sum_{t=1}^{T} \frac{v_t(i^\star)}{\sqrt{\sum_{s \le t} v_s(i^\star)}} \le 2\beta\sqrt{\sum_{t=1}^{T} v_t(i^\star)},$$

where the last inequality follows again by Orabona [2025, Lemma 4.13]. Hence, we similarly conclude that

$$\mathtt{trunc}_T(-e_{i^\star}) \le \frac{1}{\beta}\sqrt{\sum_{t=1}^{T} v_t} + \left(2\beta + \frac{1}{\beta}\right)\sqrt{\sum_{t=1}^{T} v_t(i^\star)},$$

which implies that the total cost for truncating the losses is bounded from above as

$$\mathtt{trunc}_T(p_{1:T}) + \mathtt{trunc}_T(-e_{i^\star}) \le \left(2\beta + \frac{5}{\beta}\right)\sqrt{\sum_{t=1}^{T} \bar{v}_t} + \left(2\beta + \frac{1}{\beta}\right)\sqrt{\sum_{t=1}^{T} v_t(i^\star)}.$$

**Step 2: control the regret of FTRL.** Let us now focus on the latter term in the regret decomposition, that is, the regret of FTRL on the losses $\widetilde{\ell}_1, \ldots, \widetilde{\ell}_T$. Consider any $t \in [T]$ and define

$$\widetilde{p}_{t+1} = \underset{p \in \mathbb{R}^K}{\arg\min} \left\{ \left\langle p, \widetilde{\ell}_t \right\rangle + D_{t-1}\left(p \| p_t\right) \right\}.$$

Note that for any $i \in [K]$, $\widetilde{p}_{t+1}(i) = p_t(i) \exp\left(-\eta_{t-1,i}\widetilde{\ell}_t(i)\right) \le 3p_t(i)$ by construction of $\widetilde{\ell}_t$, which is such that

$$\left|\eta_{t-1,i}\widetilde{\ell}_t(i)\right| = \eta_{t-1,i} \, |\bar{\ell}_t(i)| \frac{b_{t-1}(i)}{b_t(i)} \mathbb{1}\left(|\bar{\ell}_t(i)| \le \frac{1}{\eta_{t,i}}\right) = \eta_{t,i} \, |\bar{\ell}_t(i)| \, \mathbb{1}\left(|\bar{\ell}_t(i)| \le \frac{1}{\eta_{t,i}}\right) \le 1$$

for any $i \in [K]$ and $t > 1$, while it immediately holds for $t = 1$ since $b_0(i) = 0$ for any $i$.

Let $\varphi_t = D_{t-1}(\cdot \| p_0)$ be the regularizer used in the FTRL update. Observe that $\varphi_t$ is non-negative and twice-differentiable with Hessian having inverse $\left(\nabla^2 \varphi_t(x)\right)^{-1} = \mathrm{Diag}(\eta_{t-1} \odot x)$, and that $\varphi_{t+1}(x) \ge \varphi_t(x)$ for all $x \in \mathbb{R}^K_{\ge 0}$. Then, by standard results on FTRL with time-varying regularizers $\varphi_1, \ldots, \varphi_{T+1}$ (e.g., see Orabona [2025, Lemma 7.16]), we obtain

$$\widetilde{R}_T(e_{i^\star}) \le \varphi_{T+1}(e_{i^\star}) + \frac{1}{2} \sum_{t=1}^{T} \left\|\widetilde{\ell}_t\right\|^2_{(\nabla^2 \varphi_t(z_t))^{-1}} \tag{10}$$

for some point $z_t$ on the line segment between $p_t$ and $\widetilde{p}_{t+1}$; this also follows by the monotonicity of $\varphi_t$, i.e., $\varphi_t(p_{t+1}) \le \varphi_{t+1}(p_{t+1})$. The point $z_t$ is such that $z_t(i) \le \max\{p_t(i), \widetilde{p}_{t+1}(i)\} \le 3p_t(i)$ for all $i \in [K]$. Therefore, given the definition of the local norm, the second term of Equation (10)

satisfies

$$\frac{1}{2}\sum_{t=1}^{T}\left\|\widetilde{\ell}_t\right\|^2_{(\nabla^2\varphi_t(z_t))^{-1}} = \frac{1}{2}\sum_{t=1}^{T}\sum_{i=1}^{K}\eta_{t-1,i}z_t(i)\widetilde{\ell}_t(i)^2$$

$$\leq \frac{3}{2}\sum_{t=1}^{T}\sum_{i=1}^{K}\eta_{t-1,i}p_t(i)\widetilde{\ell}_t(i)^2 \qquad \text{(using } z_t(i) \leq 3p_t(i)\text{)}$$

$$\leq \frac{3}{2}\sum_{t=1}^{T}\sum_{i=1}^{K}\eta_{t-1,i}p_t(i)\bar{\ell}_t(i)^2 \cdot \frac{b_{t-1}(i)^2}{b_t(i)^2} \qquad \text{(using } |\widetilde{\ell}_t(i)| \leq \frac{b_{t-1}(i)}{b_t(i)}|\bar{\ell}_t(i)|\text{)}$$

$$= \frac{3\beta}{2}\sum_{t=1}^{T}\sum_{i=1}^{K}p_t(i)\bar{\ell}_t(i)^2 \cdot \frac{b_{t-1}(i)}{b_t(i)^2} \qquad \text{(definition of } \eta_{t-1,i}\text{)}$$

$$\leq \frac{3\beta}{2}\sum_{t=1}^{T}\sum_{i=1}^{K}\frac{p_t(i)\bar{\ell}_t(i)^2}{b_t(i)} \qquad \text{(using } b_{t-1}(i) \leq b_t(i)\text{)}$$

$$\leq \frac{3\beta}{2}\sum_{t=1}^{T}\frac{\bar{v}_t}{\sqrt{\sum_{s\leq t}\bar{v}_s}} \qquad \text{(definition of } b_t(i) \text{ and } \bar{v}_t\text{)}$$

$$\leq 3\beta\sqrt{\sum_{t=1}^{T}\bar{v}_t}\,,$$

where the last inequality follows again by Orabona [2025, Lemma 4.13]. On the other hand, the first term of Equation (10) is such that

$$\varphi_{T+1}(e_{i^\star}) = \frac{\log(K)-1}{\eta_{T,i^\star}} + \frac{1}{K}\sum_{i=1}^{K}\frac{1}{\eta_{T,i}}$$

$$\leq \frac{\log(K)-1}{\beta}\left(\sqrt{\sum_{t=1}^{T}v_t(i^\star)} + \sqrt{\sum_{t=1}^{T}\bar{v}_t}\right) + \frac{1}{\beta}\sqrt{\sum_{t=1}^{T}\bar{v}_t} + \frac{1}{\beta}\sum_{i=1}^{K}\frac{1}{K}\sqrt{\sum_{t=1}^{T}v_t(i)}$$

$$\leq \frac{\log(K)-1}{\beta}\sqrt{\sum_{t=1}^{T}v_t(i^\star)} + \frac{\log(K)}{\beta}\sqrt{\sum_{t=1}^{T}\bar{v}_t} + \frac{1}{\beta}\sum_{i=1}^{K}\frac{1}{K}\sqrt{\sum_{t=1}^{T}v_t(i)}\,.$$

Combining all the above results together leads to

$$\sum_{t=1}^{T}\langle p_t - e_{i^\star}, \ell_t\rangle$$

$$\leq \left(\frac{\log(K)}{\beta} + 2\beta\right)\sqrt{\sum_{t=1}^{T}v_t(i^\star)} + \left(\frac{5+\log(K)}{\beta} + 5\beta\right)\sqrt{\sum_{t=1}^{T}\bar{v}_t} + \frac{1}{\beta}\sum_{i=1}^{K}\frac{1}{K}\sqrt{\sum_{t=1}^{T}v_t(i)}\,. \;\square$$

We can now show that Theorem C.1 suffices to prove one of our main claims from Theorem 2.2. This is demonstrated by the following result.

**Corollary C.2.** *Suppose Assumption 2.1 holds. Then, Algorithm 2 with* $\beta = \sqrt{\log(K)}$ *guarantees*

$$R_T = \mathcal{O}\left(\sqrt{\theta T \log(K)}\right)\,.$$

*Proof.* Recall that, by Theorem C.1, Algorithm 2 with $\beta = \sqrt{\log(K)}$ guarantees

$$\sum_{t=1}^{T}\langle p_t - e_{i^\star}, \ell_t\rangle = \mathcal{O}\left(\sqrt{\log(K)\left(\sum_{t=1}^{T}\bar{v}_t + \sum_{t=1}^{T}v_t(i^\star)\right)} + \frac{1}{K}\sum_{i=1}^{K}\sqrt{\sum_{t=1}^{T}v_t(i)}\right) \qquad (11)$$

for any sequence of losses. Additionally, in a similar way as in the proof of Corollary B.3, we have that $\mathbb{E}_t[\bar{v}_t] \leq \theta$ and that $\mathbb{E}_t[v_t(i)] \leq 4\theta$ for any $i \in [K]$, under Assumption 2.1.

We can analogously observe that $R_T = \max_{i^\star \in [K]} \mathbb{E}[\langle p_t - e_{i^\star}, \ell_t \rangle]$ or, in other words, that $R_T$ essentially corresponds to the expectation of the left-hand side of Equation (11). Then, we consider the expectation of its right-hand side and, by applying Jensen's inequality with respect to the square root and the tower rule of expectation, we can finally show that

$$\mathbb{E}\left[ \sqrt{\log(K) \sum_{t=1}^{T} (\bar{v}_t + v_t(i^\star))} + \frac{1}{K} \sum_{i=1}^{K} \sqrt{\sum_{t=1}^{T} v_t(i)} \right]$$

$$\leq \sqrt{\log(K) \, \mathbb{E}\left[ \sum_{t=1}^{T} \big( \mathbb{E}_t[\bar{v}_t] + \mathbb{E}_t[v_t(i^\star)] \big) \right]} + \frac{1}{K} \sum_{i=1}^{K} \sqrt{\mathbb{E}\left[ \sum_{t=1}^{T} \mathbb{E}_t[v_t(i)] \right]}$$

$$\leq C \sqrt{\theta T \log(K)}$$

for some constant $C > 0$. This concludes the proof. $\qquad \square$

## C.1   Computing the Update in Algorithm 2

We briefly discuss the update defining $p_{t+1}$ in Algorithm 2. For any $\eta \in \mathbb{R}_{>0}^{K}$, consider an optimization problem of the form $\inf_{p \in \mathcal{P}} \langle p, L \rangle + D(p\|q)$, where $L \in \mathbb{R}^{K}$, $q \in \mathcal{P}$, and

$$D(p\|q) = \sum_{i=1}^{K} \frac{1}{\eta_i} \left[ p(i) \log\left( \frac{p(i)}{q(i)} \right) + p(i) - q(i) \right].$$

As the probability simplex is compact and the mapping $p \mapsto \langle p, L \rangle + D(p\|q)$ is continuous, the infimum is attained at some $p^\star \in \mathcal{P}$. The Lagrangian $\mathcal{L}(p, \lambda)$ of the optimization problem is defined for $p \in \mathbb{R}_{\geq 0}^{K}$ and $\lambda \in \mathbb{R}$ as

$$\mathcal{L}(p, \lambda) = \langle p, L \rangle + \sum_{i=1}^{K} \frac{1}{\eta_i} \left[ p(i) \log\left( \frac{p(i)}{q(i)} \right) - p(i) + q(i) \right] + \lambda \left( \langle p, \mathbb{1} \rangle - 1 \right).$$

For any $j \in [K]$, differentiate with respect to $p(j)$ to get

$$\frac{\partial \mathcal{L}}{\partial p(j)}(p, \lambda) = L(j) + \frac{1}{\eta_j} \left[ \log\left( \frac{p(j)}{q(j)} \right) + 1 - 1 \right] + \lambda = L(j) + \lambda + \frac{1}{\eta_j} \log\left( \frac{p(j)}{q(j)} \right).$$

Setting it to zero, we get

$$p^\star(j) = q(j) \exp\left( -\eta_j [L(j) + \lambda] \right).$$

One then wants to find the value of $\lambda$ by enforcing the constraint on $p^\star$, namely $\langle p^\star, \mathbb{1} \rangle = 1$, which gives the following condition:

$$\sum_{i=1}^{K} q(i) \exp\left( -\eta_i [L(i) + \lambda] \right) = 1.$$

If the learning rate did not depend on the coordinate, we could take the term that depends on $\lambda$ out of the sum to get a closed-form solution. Plugging it back into $p^\star$ would give a softmax distribution, but this is not possible here. Instead, one can efficiently compute the normalization constant $\lambda$ with a line-search.

# D   Technical Results for Section 5.3

## D.1   Proof of Theorem 2.5

*Proof.* We start with some useful inequalities. Let $a, b, y_t \in \mathbb{R}$ and let $\bar{\ell}_t = -r_t$ for any $t \in [T]$. By the 2-strong convexity of the function $x \mapsto (x - y_t)^2$, we have

$$\left( \frac{a+b}{2} - y_t \right)^2 \leq \frac{1}{2}(a - y_t)^2 + \frac{1}{2}(b - y_t)^2 - \frac{(a-b)^2}{8}. \tag{12}$$

---

**Algorithm 3** `LoOT-Free OMD` for the Squared Loss

---

**Inputs:** Number of experts $K \geq 2$, minimum mass coefficient $\alpha \in (0, 1]$, learning rate coefficient $\beta > 0$.
**Initialize:** $p_1(i) \leftarrow 1/K$ for all $i \in [K]$.
**for** $t = 1, \ldots, T$ **do**
   Receive $\mathbf{z}_{t,i}$ for all $i \in [K]$.
   Set $\bar{\mathbf{z}}_t \leftarrow \sum_{i=1}^K p_t(i) \mathbf{z}_{t,i}$.
   Observe $y_t$.
   Set $\ell_t(i) \leftarrow \left( \mathbf{z}_{t,i}(\bar{\mathbf{z}}_t - y_t) + \frac{1}{2}(\mathbf{z}_{t,i} - y_t)^2 \right)$.
   Set $r_t(i) \leftarrow \langle \ell_t, p_t \rangle - \ell_t(i)$ for all $i \in [K]$.
   Set $v_t(i) \leftarrow r_t(i)^2$ for all $i \in [K]$.
   Set $\bar{v}_t \leftarrow \sum_{i=1}^K p_t(i) v_t(i)$.
   Set $\eta_{t,i} \leftarrow \beta \max \left\{ \sum_{s \leq t} \bar{v}_s, \sum_{s \leq t} v_s(i) \right\}^{-1/2}$ for all $i \in [K]$.
   Set $\widetilde{\ell}_t(i) \leftarrow -r_t(i) \mathbb{1}\left( |r_t(i)| \leq 1/\eta_{t,i} \right)$ for all $i \in [K]$.
   Set $p_{t+1} \leftarrow \arg\min_{p \in \mathcal{P}_\alpha} \langle p, \widetilde{\ell}_t \rangle + D_t(p \| p_t)$.
**end for**

---

Furthermore, we also have that $(a - y_t)^2 - (b - y_t)^2 = 2(y_t - a)(b - a) - (a - b)^2$, sometimes referred to as a polarization identity (here, for the Euclidean norm on $\mathbb{R}$). Moving to the regret analysis, let us denote $i^\star = \arg\min_{i \in [K]} \mathbb{E}\left[ \sum_{t=1}^T (y_t - \mathbf{z}_{t,i})^2 \right]$. Splitting the sum in halves and rearranging the terms using that $p_t$ is a probability distribution over $[K]$, the loss of the learner at time $t$ can be bounded from above as

$$
\begin{aligned}
(\bar{\mathbf{z}}_t - y_t)^2 &= \left( \left\langle p_t, \frac{1}{2} \mathbf{z}_t \right\rangle + \frac{1}{2} \bar{\mathbf{z}}_t - y_t \right)^2 \\
&= \left( \sum_{i=1}^K p_t(i) \left[ \frac{1}{2} \mathbf{z}_{t,i} + \frac{1}{2} \bar{\mathbf{z}}_t - y_t \right] \right)^2 \\
&\leq \sum_{i=1}^K p_t(i) \left[ \frac{1}{2} \mathbf{z}_{t,i} + \frac{1}{2} \bar{\mathbf{z}}_t - y_t \right]^2 ,
\end{aligned}
$$

where the inequality follows from Jensen's inequality. Applying Equation (12) for any $i \in [K]$ with $a = \mathbf{z}_{t,i}$ and $b = \bar{\mathbf{z}}_t$, and again using $\langle p_t, \mathbb{1} \rangle = 1$, we further get

$$
\begin{aligned}
(\bar{\mathbf{z}}_t - y_t)^2 &\leq \frac{1}{2} \sum_{i=1}^K p_t(i)(\mathbf{z}_{t,i} - y_t)^2 + \frac{1}{2}(\bar{\mathbf{z}}_t - y_t)^2 - \frac{1}{8} \sum_{i=1}^K p_t(i)(\mathbf{z}_{t,i} - \bar{\mathbf{z}}_t)^2 \\
&= \frac{1}{2} \sum_{i=1}^K p_t(i)(\mathbf{z}_{t,i} - y_t)^2 + \underbrace{\frac{1}{2}(\bar{\mathbf{z}}_t - y_t)^2 - \frac{1}{2}(\mathbf{z}_{t,i^\star} - y_t)^2}_{(\Diamond)} \\
&\quad - \frac{1}{8} \sum_{i=1}^K p_t(i)(\mathbf{z}_{t,i} - \bar{\mathbf{z}}_t)^2 + (\mathbf{z}_{t,i^\star} - y_t)^2 - \frac{1}{2}(\mathbf{z}_{t,i^\star} - y_t)^2 .
\end{aligned}
$$

Using the polarization identity on $(\Diamond)$ with $a = \bar{\mathbf{z}}_t$ and $b = \mathbf{z}_{t,i^\star}$ leads to

$$
\begin{aligned}
(\bar{\mathbf{z}}_t - y_t)^2 &\leq \frac{1}{2} \sum_{i=1}^K p_t(i)(\mathbf{z}_{t,i} - y_t)^2 + (y_t - \bar{\mathbf{z}}_t)(\mathbf{z}_{t,i^\star} - \bar{\mathbf{z}}_t) - \frac{1}{2}(\bar{\mathbf{z}}_t - \mathbf{z}_{t,i^\star})^2 \\
&\quad - \frac{1}{8} \sum_{i=1}^K p_t(i)(\mathbf{z}_{t,i} - \bar{\mathbf{z}}_t)^2 + (\mathbf{z}_{t,i^\star} - y_t)^2 - \frac{1}{2}(\mathbf{z}_{t,i^\star} - y_t)^2 .
\end{aligned}
$$

We recall that Algorithm 3 uses the loss $\ell_t(i) = \frac{1}{2}(\mathbf{z}_{t,i} - y_t)^2 + \mathbf{z}_{t,i}(\bar{\mathbf{z}}_t - y_t)$ for any $i \in [K]$. Rearranging the first two terms in the upper bound, we have

$$(\bar{\mathbf{z}}_t - y_t)^2 \leq \langle p_t, \ell_t \rangle - \underbrace{\mathbf{z}_{t,i^\star}(\bar{\mathbf{z}}_t - y_t)}_{(\clubsuit)} - \frac{1}{2}(\bar{\mathbf{z}}_t - \mathbf{z}_{t,i^\star})^2$$

$$- \frac{1}{8}\sum_{i=1}^{K} p_t(i)(\mathbf{z}_{t,i} - \bar{\mathbf{z}}_t)^2 + (\mathbf{z}_{t,i^\star} - y_t)^2 - \underbrace{\frac{1}{2}(\mathbf{z}_{t,i^\star} - y_t)^2}_{(\spadesuit)}$$

$$= \langle p_t, \ell_t \rangle - \ell_t(i^\star) - \frac{1}{2}(\bar{\mathbf{z}}_t - \mathbf{z}_{t,i^\star})^2 \qquad (\text{using } (\clubsuit) + (\spadesuit) = \ell_t(i^\star))$$

$$- \frac{1}{8}\sum_{i=1}^{K} p_t(i)(\mathbf{z}_{t,i} - \bar{\mathbf{z}}_t)^2 + (\mathbf{z}_{t,i^\star} - y_t)^2 .$$

At this point, we can move the last term in the right-hand side to the left-hand side to obtain

$$(\bar{\mathbf{z}}_t - y_t)^2 - (\mathbf{z}_{t,i^\star} - y_t)^2 \leq (\langle p_t, \ell_t \rangle - \ell_t(i^\star)) - \frac{1}{2}(\bar{\mathbf{z}}_t - \mathbf{z}_{t,i^\star})^2 - \frac{1}{8}\sum_{i=1}^{K} p_t(i)(\mathbf{z}_{t,i} - \bar{\mathbf{z}}_t)^2 . \quad (13)$$

Now, from Theorem B.2 with $\alpha = \frac{1}{T}$ and $\beta = \sqrt{\log(KT)}$ we have

$$\sum_{t=1}^{T}(\langle p_t, \ell_t \rangle - \ell_t(i^\star)) \leq 11\sqrt{\log(KT)}\left(\sqrt{\sum_{t=1}^{T}\bar{v}_t} + \sqrt{\sum_{t=1}^{T}v_t(i^\star)}\right) . \qquad (14)$$

We continue by bounding $\bar{v}_t = \sum_{i=1}^{K} p_t(i)(\ell_t(i) - \langle p_t, \ell_t \rangle)^2$ from above. By definition of the loss $\ell_t$, the square inside the sum can be bounded, for any $i \in [K]$, by

$$(\ell_t(i) - \langle p_t, \ell_t \rangle)^2 = \left(\frac{1}{2}\left[(\mathbf{z}_{t,i} - y_t)^2 - \langle p_t, (\mathbf{z}_t - y_t \cdot \mathbb{1})^2 \rangle\right] + (\mathbf{z}_{t,i} - \bar{\mathbf{z}}_t)(\bar{\mathbf{z}}_t - y_t)\right)^2$$

$$\leq 2\left[(\mathbf{z}_{t,i} - \bar{\mathbf{z}}_t)(\bar{\mathbf{z}}_t - y_t)\right]^2 + \frac{1}{2}\left[(\mathbf{z}_{t,i} - y_t)^2 - \langle p_t, (\mathbf{z}_t - y_t \cdot \mathbb{1})^2 \rangle\right]^2 , \quad (15)$$

where we used the inequality $(a + b)^2 \leq 2a^2 + 2b^2$ for any $a, b \in \mathbb{R}$. Plugging it into the definition of $\bar{v}_t$, this gives

$$\bar{v}_t \leq 2(\bar{\mathbf{z}}_t - y_t)^2\sum_{i=1}^{K} p_t(i)[\mathbf{z}_{t,i} - \bar{\mathbf{z}}_t]^2 + \frac{1}{2}\sum_{i=1}^{K} p_t(i)\left((\mathbf{z}_{t,i} - y_t)^2 - \sum_{j=1}^{K} p_t(j)(\mathbf{z}_{t,j} - y_t)^2\right)^2 .$$

We continue by bounding from above the second term on the right-hand side. The difference inside the square can be equivalently rewritten as

$$(\mathbf{z}_{t,i} - y_t)^2 - \sum_{j=1}^{K} p_t(j)(\mathbf{z}_{t,j} - y_t)^2$$

$$= [(\mathbf{z}_{t,i} - \bar{\mathbf{z}}_t) - (y_t - \bar{\mathbf{z}}_t)]^2 - \sum_{j=1}^{K} p_t(j)[(\mathbf{z}_{t,j} - \bar{\mathbf{z}}_t) - (y_t - \bar{\mathbf{z}}_t)]^2$$

$$= (\mathbf{z}_{t,i} - \bar{\mathbf{z}}_t)^2 - 2(y_t - \bar{\mathbf{z}}_t)(\mathbf{z}_{t,i} - \bar{\mathbf{z}}_t) + (y_t - \bar{\mathbf{z}}_t)^2 - \sum_{j=1}^{K} p_t(j)(\mathbf{z}_{t,j} - \bar{\mathbf{z}}_t)^2$$

$$+ 2(y_t - \bar{\mathbf{z}}_t)\sum_{j=1}^{K} p_t(j)(\mathbf{z}_{t,j} - \bar{\mathbf{z}}_t) - (y_t - \bar{\mathbf{z}}_t)^2 \qquad (\langle p_t, \mathbb{1} \rangle = 1)$$

$$= (\mathbf{z}_{t,i} - \bar{\mathbf{z}}_t)^2 - 2(y_t - \bar{\mathbf{z}}_t)(\mathbf{z}_{t,i} - \bar{\mathbf{z}}_t) - \sum_{j=1}^{K} p_t(j)(\mathbf{z}_{t,j} - \bar{\mathbf{z}}_t)^2 , \qquad (16)$$

where the last equality is by definition of $\bar{\mathbf{z}}_t$. We plug it back in the sum above, and using again the inequality $(a+b)^2 \leq 2a^2 + 2b^2$, we get

$$\sum_{i=1}^{K} p_t(i) \left( (\mathbf{z}_{t,i} - y_t)^2 - \sum_{j=1}^{K} p_t(j)(\mathbf{z}_{t,j} - y_t)^2 \right)^2$$

$$\leq 2 \sum_{i=1}^{K} p_t(i)(\mathbf{z}_{t,i} - \bar{\mathbf{z}}_t)^4 + 2 \sum_{i=1}^{K} p_t(i) \left[ 2(y_t - \bar{\mathbf{z}}_t)(\mathbf{z}_{t,i} - \bar{\mathbf{z}}_t) + \sum_{j=1}^{K} p_t(j)(\mathbf{z}_{t,j} - \bar{\mathbf{z}}_t)^2 \right]^2 .$$

Expanding the square in the second term, we get that the cross-product is equal to zero by definition of $\bar{\mathbf{z}}_t$, thus

$$\sum_{i=1}^{K} p_t(i) \left( (\mathbf{z}_{t,i} - y_t)^2 - \sum_{j=1}^{K} p_t(j)(\mathbf{z}_{t,j} - y_t)^2 \right)^2$$

$$\leq 2 \sum_{i=1}^{K} p_t(i)(\mathbf{z}_{t,i} - \bar{\mathbf{z}}_t)^4 + 8(y_t - \bar{\mathbf{z}}_t)^2 \sum_{i=1}^{K} p_t(i)(\mathbf{z}_{t,i} - \bar{\mathbf{z}}_t)^2 + 2 \left[ \sum_{j=1}^{K} p_t(j)(\mathbf{z}_{t,j} - \bar{\mathbf{z}}_t)^2 \right]^2 ,$$

where we used $\langle p_t, \mathbb{1} \rangle = 1$. Using Jensen's inequality,

$$\sum_{i=1}^{K} p_t(i) \left( (\mathbf{z}_{t,i} - y_t)^2 - \sum_{j=1}^{K} p_t(j)(\mathbf{z}_{t,j} - y_t)^2 \right)^2$$

$$\leq 4 \sum_{i=1}^{K} p_t(i)(\mathbf{z}_{t,i} - \bar{\mathbf{z}}_t)^4 + 8(y_t - \bar{\mathbf{z}}_t)^2 \sum_{i=1}^{K} p_t(i)(\mathbf{z}_{t,i} - \bar{\mathbf{z}}_t)^2 .$$

Plugging it back into the upper-bound on $\bar{v}_t$, we obtain

$$\bar{v}_t \leq 6(\bar{\mathbf{z}}_t - y_t)^2 \sum_{i=1}^{K} p_t(i)[\mathbf{z}_{t,i} - \bar{\mathbf{z}}_t]^2 + 2 \sum_{i=1}^{K} p_t(i)(\mathbf{z}_{t,i} - \bar{\mathbf{z}}_t)^4 .$$

Using $\mathbb{E}_t[y_t^2] \leq \sigma$ and $|\mathbf{z}_{t,i}| \leq Y$, we find that

$$\mathbb{E}_t[\bar{v}_t] \leq (20Y^2 + 12\sigma) \sum_{i=1}^{K} p_t(i)(\mathbf{z}_{t,i} - \bar{\mathbf{z}}_t)^2 . \tag{17}$$

We now bound $v_t(i)$ for any given $i \in [K]$. As with $\bar{v}_t$, Equation (15) gives

$$(\ell_t(i) - \langle p_t, \ell_t \rangle)^2 \leq 2(\bar{\mathbf{z}}_t - y_t)^2(\mathbf{z}_{t,i} - \bar{\mathbf{z}}_t)^2 + \frac{1}{2} \left[ (\mathbf{z}_{t,i} - y_t)^2 - \sum_{j=1}^{K} p_t(j)(\mathbf{z}_{t,j} - y_t)^2 \right]^2 .$$

Reusing Equation (16), we can write the second term on the right-hand-side as

$$\left( (\mathbf{z}_{t,i} - y_t)^2 - \sum_{j=1}^{K} p_t(j)(\mathbf{z}_{t,j} - y_t)^2 \right)^2$$

$$= \left( (\mathbf{z}_{t,i} - \bar{\mathbf{z}}_t)^2 - 2(y_t - \bar{\mathbf{z}}_t)(\mathbf{z}_{t,i} - \bar{\mathbf{z}}_t) - \sum_{j=1}^{K} p_t(j)(\mathbf{z}_{t,j} - \bar{\mathbf{z}}_t)^2 \right)^2$$

$$\leq \left[ 3(\mathbf{z}_{t,i} - \bar{\mathbf{z}}_t)^2 + 6(y_t - \bar{\mathbf{z}}_t)^2 \right](\mathbf{z}_{t,i} - \bar{\mathbf{z}}_t)^2 + 3 \left( \sum_{j=1}^{K} p_t(j)(\mathbf{z}_{t,j} - \bar{\mathbf{z}}_t)^2 \right)^2$$

$$\leq \left[ 3(\mathbf{z}_{t,i} - \bar{\mathbf{z}}_t)^2 + 6(y_t - \bar{\mathbf{z}}_t)^2 \right](\mathbf{z}_{t,i} - \bar{\mathbf{z}}_t)^2 + 3 \left( \max_{k \in [K]} (\mathbf{z}_{t,k} - \bar{\mathbf{z}}_t)^2 \right) \sum_{j=1}^{K} p_t(j)(\mathbf{z}_{t,j} - \bar{\mathbf{z}}_t)^2 ,$$

where the first inequality follows from $(a + b + c)^2 \leq 3a^2 + 3b^2 + 3c^2$ valid for any $a, b, c \in \mathbb{R}$ and the second inequality follows from Jensen's inequality. Thus, using $\mathbb{E}_t\left[y_t^2\right] \leq \sigma$ and $|\mathbf{z}_{t,i}| \leq Y$, we find that

$$\mathbb{E}_t\left[v_t\left(i\right)\right] \leq \left(16Y^2 + 10\sigma\right)\left(\mathbf{z}_{t,i} - \bar{\mathbf{z}}_t\right)^2 + 6Y^2 \sum_{i=1}^{K} p_t\left(i\right)\left(\mathbf{z}_{t,i} - \bar{\mathbf{z}}_t\right)^2 . \tag{18}$$

Consider now any $\gamma > 0$. By using the bounds on $\mathbb{E}_t\left[v_t\left(i\right)\right]$ and $\mathbb{E}_t\left[\bar{v}_t\right]$, together with Equations (13) and (14), we finally find that

$$\mathbb{E}\left[\sum_{t=1}^{T}\left(\left(\bar{\mathbf{z}}_t - y_t\right)^2 - \left(\mathbf{z}_{t,i^\star} - y_t\right)^2\right)\right]$$

$$\leq \mathbb{E}\left[11\sqrt{\log\left(KT\right)}\left(\sqrt{\sum_{t=1}^{T}\bar{v}_t} + \sqrt{\sum_{t=1}^{T}v_t\left(i^\star\right)}\right)\right]$$

$$- \mathbb{E}\left[\sum_{t=1}^{T}\left(\frac{1}{2}\left(\mathbf{z}_{t,i^\star} - \bar{\mathbf{z}}_t\right)^2 + \frac{1}{8}\sum_{j=1}^{K} p_t\left(j\right)\left(\mathbf{z}_{t,j} - \bar{\mathbf{z}}_t\right)^2\right)\right] \qquad \text{(Equations (13) and (14))}$$

$$\leq \mathbb{E}\left[11\sqrt{2\log\left(KT\right)}\left(\sqrt{\sum_{t=1}^{T}\bar{v}_t + \sum_{t=1}^{T}v_t\left(i^\star\right)}\right)\right]$$

$$- \mathbb{E}\left[\sum_{t=1}^{T}\left(\frac{1}{2}\left(\mathbf{z}_{t,i^\star} - \bar{\mathbf{z}}_t\right)^2 + \frac{1}{8}\sum_{j=1}^{K} p_t\left(j\right)\left(\mathbf{z}_{t,j} - \bar{\mathbf{z}}_t\right)^2\right)\right]$$

$$\leq \frac{121\log\left(KT\right)}{\gamma} - \mathbb{E}\left[\sum_{t=1}^{T}\left(\frac{1}{2}\left(\mathbf{z}_{t,i^\star} - \bar{\mathbf{z}}_t\right)^2 + \frac{1}{8}\sum_{j=1}^{K} p_t\left(j\right)\left(\mathbf{z}_{t,j} - \bar{\mathbf{z}}_t\right)^2\right)\right]$$

$$+ \gamma\mathbb{E}\left[\left(8Y^2 + 5\sigma\right)\sum_{t=1}^{T}\left(\mathbf{z}_{t,i^\star} - \bar{\mathbf{z}}_t\right)^2 + \left(13Y^2 + 6\sigma\right)\sum_{t=1}^{T}\sum_{i=1}^{K} p_t\left(i\right)\left(\mathbf{z}_{t,i} - \bar{\mathbf{z}}_t\right)^2\right]$$

$$\leq \frac{121\log\left(KT\right)}{\gamma} - \mathbb{E}\left[\sum_{t=1}^{T}\left(\frac{1}{2}\left(\mathbf{z}_{t,i^\star} - \bar{\mathbf{z}}_t\right)^2 + \frac{1}{8}\sum_{j=1}^{K} p_t\left(j\right)\left(\mathbf{z}_{t,j} - \bar{\mathbf{z}}_t\right)^2\right)\right]$$

$$+ \gamma\left(104Y^2 + 48\sigma\right)\mathbb{E}\left[\frac{1}{2}\sum_{t=1}^{T}\left(\mathbf{z}_{t,i^\star} - \bar{\mathbf{z}}_t\right)^2 + \frac{1}{8}\sum_{t=1}^{T}\sum_{i=1}^{K} p_t\left(i\right)\left(\mathbf{z}_{t,i} - \bar{\mathbf{z}}_t\right)^2\right]$$

$$= C\left(Y^2 + \sigma\right)\log\left(KT\right)$$

for a sufficiently large constant $C > 0$, where the third inequality follows from Equations (17) and (18) together with the fact that $\sqrt{ab} = \inf_{\gamma > 0}\frac{1}{2\gamma}a + \frac{\gamma}{2}b$ for any $a, b \geq 0$, whereas the last inequality follows by the choice of $\gamma = \left(104Y^2 + 48\sigma\right)^{-1}$. This completes the proof. $\qquad\square$

## E  Comparison with Gökcesu and Kozat [2022]

In this section, we provide further details about the comparison with Gökcesu and Kozat [2022]. The first regret guarantees we compare against are those provided by their Theorem IV.7 and Theorem V.2. One may immediately observe that those regret bounds present an additive term $E_T$ (multiplied by a logarithmic factor) equivalent to our $M_T = \max_{t,i}|r_t(i)|$, for which we already prove that the $\sqrt{KT}$ lower bound holds even with i.i.d. losses.

Hence, the main comparison is mainly with respect to Corollary IV.8 and Corollary V.3 in Gökcesu and Kozat [2022]. The proof of Corollary IV.8 and Corollary V.3 rely on their Lemma IV.1, which requires that, in the notation of Gökcesu and Kozat [2022], $\eta_t|l_{t,m} - \mu_t| \leq 1$ for all $t$ and all $m$

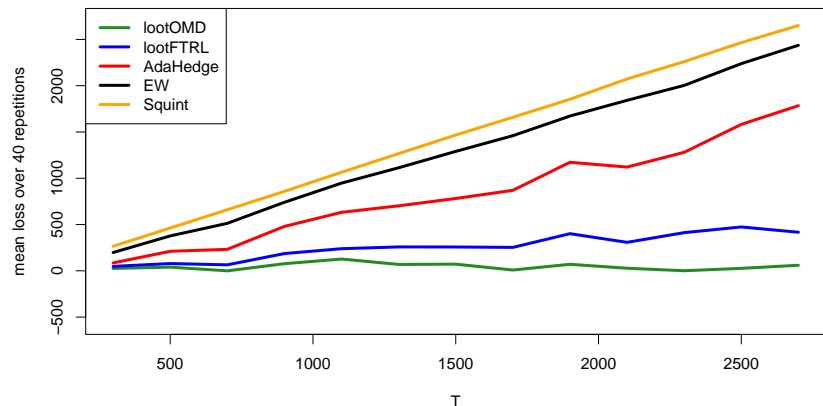

Figure 1: Results of experiments with heavy-tailed losses.

Figure 2: Results of experiments with heavy-tailed losses. Dotted lines represent mean $\pm$ one standard deviation.

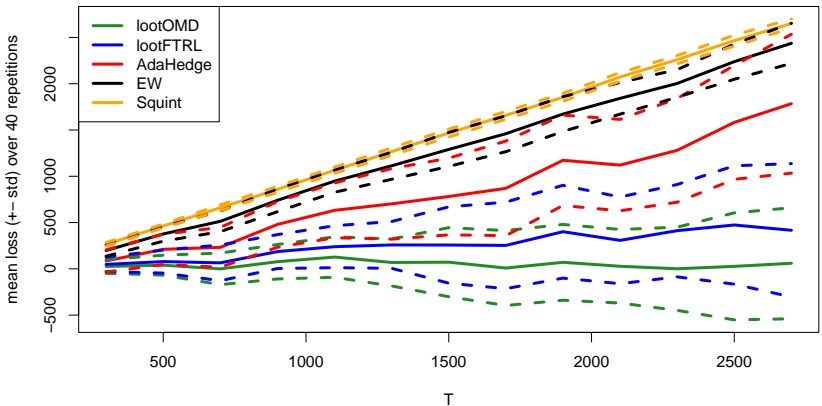

for Equation (b) in the proof of Lemma IV.1 to be true. However, with the learning rates given in Corollary IV.8 and Corollary V.3, $\eta_t|l_{t,m} - \mu_t| \leq 1$ does not hold. In Corollary IV.8 they choose $\eta_t \geq \sqrt{\frac{1}{\sum_{s=1}^t \mathbb{E}_{p_s}[(l_{s,m} - \mu_s)^2]}}$, which with $\mu_s = \min_m l_{s,m}$ for $t = 1$ with $l_{1,m} = 0$ if $m \neq K$ and $l_{1,K} = 1$, can be seen to lead to $\eta_t|l_{t,m} - \mu_t| \geq \sqrt{K} > 1$. With Corollary V.3, we run into a similar issue. We do not see a way to fix these issues.

## F Details on the Experiments

In this section we provide details on the experiments. All experiments were run on a Macbook Air with 8GB of RAM and an Apple M2 processor. We ran two sets of experiments, and the experiments in each set were run for $K \in \{15, 25, \ldots, 135\}$. For each instance we set $T = 20K$.

In the first set of experiments the losses mimic the construction we used in Section 4. The expert losses were equal to $\ell_t(i) = \mathbb{1}[i \neq 1] + \varepsilon_{t,i}$, where $\varepsilon_{t,i} = \zeta_{t,i} X_{t,i}$ with $\zeta_{t,i}$ a Rademacher random variable and

$$X_{t,i} = \begin{cases} 0 & \text{w.p. } 1 - 1/T \\ \sqrt{KT} & \text{w.p. } 1/T \end{cases}.$$

Figure 3: Results of experiments with non-heavy-tailed losses.

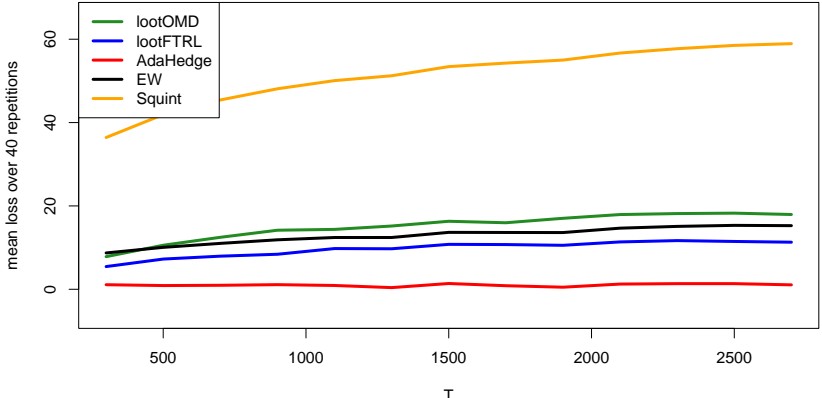

Figure 4: Results of experiments with non-heavy-tailed losses. Dotted lines represent mean $\pm$ one standard deviation.

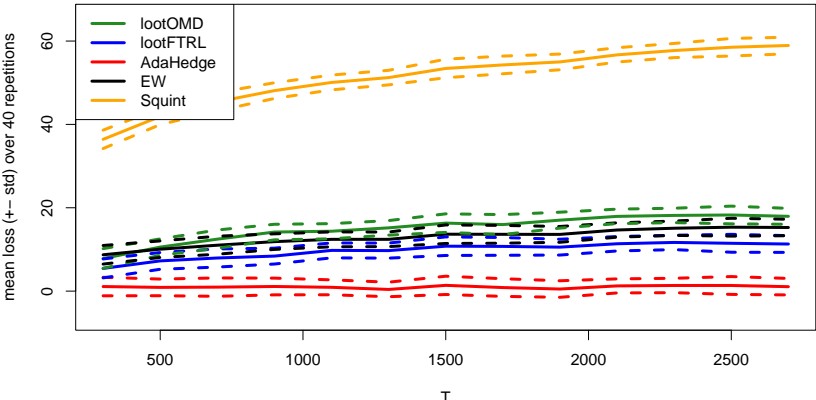

In the second set of experiments we use a similar construction, where the losses were equal to $\ell_t(i) = \mathbb{1}[i \neq 1] + \varepsilon_{t,i}$, where $\varepsilon_{t,i} = \zeta_{t,i}\widetilde{X}_{t,i}$ with $\zeta_{t,i}$ a Rademacher random variable and

$$\widetilde{X}_{t,i} = \begin{cases} 0 & \text{w.p. } 1 - 1/T \\ 2 & \text{w.p. } 1/T \end{cases}.$$

The algorithms we implemented were Squint with the improper prior [Koolen and Van Erven, 2015], AdaHedge [De Rooij et al., 2014], and an instance of Exponential Weights (EW) algorithm akin to the algorithm of [Cesa-Bianchi et al., 2007]. Specifically, we ran the FTRL version of exponential weights on the instantaneous regrets with learning rate $\eta_t = \min\left\{ \frac{1}{\max_{t,i}|\ell_t(i)|}, \sqrt{\frac{\log(K)}{\sum_{s \leq t} \bar{v}_t}} \right\}$. We gave this instance of EW the maximum loss, otherwise it would not provide any guarantees. We could have also opted for a doubling trick, but this is known to deteriorate performance. Likewise, we gave Squint the value of $\max_{t,i}|\ell_t(i)|$, as this is a required parameter for Squint. The algorithm of [Mhammedi et al., 2019] could have instead been used to learn $\max_{t,i}|\ell_t(i)|$ online, but seeing that a similar idea as the doubling trick is part of their algorithm, we suspect this would only deteriorate performance.

In the first set of experiments we expect Squint, AdaHedge, and EW to perform poorly due to the issues described in Section 4. In the second set of experiments we expect similar behaviour from all algorithms.

As can be seen from the results in Figure 1, algorithms not tailored to adapt to $\theta$ fare considerably worse in the heavy-tailed loss setting we consider. We therefore conclude that the lower-order terms in the regret bounds of these algorithms are not an artefact of the analysis, but rather represent the problematic behaviour of these algorithms in the face of heavy-tailed losses.

The results for the second set of experiments is similarly as expected with one exception: the performance of Squint. Squint fares considerably worse than the other algorithms. However, upon inspection, it seems that the performance of Squint is still below what is predicted by theory. The regret bound of Squint contains a $15 \log(1 + K(2 + \log(T + 1)))$ term, which for the smallest values of $K, T$ is slightly larger than 71.

