# OpenReview forum: "When Lower-Order Terms Dominate: Adaptive Expert Algorithms for Heavy-Tailed Losses"
_NeurIPS.cc/2025/Conference — NeurIPS 2025 poster_

### Official Review · Reviewer_XFbG · 2025-06-24

**Clarity:** 3
**Significance:** 3
**Originality:** 3
**Rating:** 5
**Confidence:** 3

**Summary:**

This paper addresses the problem of prediction with expert advice under the assumption that losses are heavy-tailed but have bounded second moments. The authors develop adaptive algorithms that improve the dependence on the number of experts from the usual $\sqrt{K}$ to $\sqrt{\log K}$ in the regret bound, under appropriate assumptions. They consider both the standard linear (Hedge) setting and a quadratic loss variant, providing adaptive algorithms based on Online Mirror Descent (OMD) and Follow-The-Regularized-Leader (FTRL).

**Questions:**

1. Do the authors have insight into potential lower bounds for these settings? For example, do known adversarial lower bounds apply here, given that bounded adversarial losses would satisfy Assumption 2.1? If so, can the optimality of established rates be discussed?

2. Are the values on $Y$-axis of the plots regret?

**Ethical Concerns:**

["NO or VERY MINOR ethics concerns only"]

**Final Justification:**

I have initial concerns about the motivation for the problem. However, the authors have addressed my concerns and I am happy with the current score.

**Limitations:**

yes

**Quality:**

3

**Strengths And Weaknesses:**

The expert setting with heavy-tailed losses is relevant in practical scenarios involving outliers, noise, or mechanisms like local differential privacy. The paper makes an insightful observation that terms traditionally treated as lower-order can, in fact, dominate the regret when only second-moment bounds are available.

From a technical standpoint, the paper is solid. The proposed algorithms are interesting variants of OMD and FTRL, and the analysis appears to be non-trivial. The authors also support their claims with synthetic experiments.

My main concern lies in the lack of discussion around the motivation and interpretability of the assumptions. For instance, Assumption 2.3 is introduced without much explanation. It would be helpful to include an intuitive explanation of $\Delta_{\min}$ and $C$, and how these constants behave under different distributions. Can $C$ grow with $T$, or is it typically constant for distributions with bounded second moments? A discussion akin to Section 4, which provided insight into why certain terms dominate, could help clarify the meaning and implications of these parameters.

Similarly, Section 2.2 introduces a quadratic loss model, but it is not clear why this setting is interesting or whether it arises naturally in known applications. Is it connected to scalar regression, or is there prior literature studying this variant? A brief contextualization would help establish its relevance.

Overall, the paper is technically strong and contributes important insights to the literature on adaptive online learning under heavy-tailed losses. However, additional exposition around the assumptions and modeling choices would strengthen the presentation.

---

> ### Author Rebuttal · Authors · 2025-07-30
>
> Thank you for taking the time to review our paper, for the positive feedback, and for recognizing our technical work. We address your points below.
>
> **Regarding the interpretability of Assumption 2.3**. This assumption is intended to model environments that are not fully adversarial but have some underlying stochastic structure. Intuitively, $\Delta_{\min}$ represents the minimum performance gap between the best expert and any other expert, and the constant $C$ acts as a "corruption" budget, allowing for some adversarial deviations from this stochastic structure. In a purely i.i.d. setting, we would have $C = 0$. The question of whether $C$ can grow with $T$ or $K$ is a good one: In the literature (e.g., [5, 6]), $C$ is typically assumed to be a constant. However, like the algorithm of Zimmert and Seldin [5], our algorithm can also tolerate corruptions that are at most logarithmic and still have meaningful guarantees. To the best of our knowledge, it is an open problem whether it is possible to tolerate a linear amount of corruption (see, e.g., Section 5.2 of [5]). See also our answer to Reviewer ke8c for additional details on Assumption 2.3.
>
> **Regarding the motivation for the heavy-tailed setting**. We refer to our answer to Reviewer fiH4 for a few motivating examples.
>
> **Regarding the squared loss**. Indeed, our main motivation lies in scalar regression. This framework enables us to generalize the classical results of Vovk [1] to the more challenging setting where the outcomes $y_t$ can be heavy-tailed. This is an important setting with many applications, such as regression under heavy-tailed distributions, a topic surveyed by Lugosi and Mendelson [2, Section 5]. When we have different models to use but do not know which one is the best, our algorithm can learn (online) to perform nearly as well as this single best model, all without restrictive assumptions on problem parameters like knowledge of $\theta$.
>
> **(Q1) Regarding the lower bounds**. Yes, this is a great question. The standard adversarial lower bounds for the experts problem apply directly to our setting because, as you pointed out, the setting with bounded losses is a special case of our Assumption 2.1. Therefore, the $\Omega(\sqrt{T \log K})$ lower bound (see, e.g., Theorem 3.6 from Cesa-Bianchi and Lugosi [3], or Theorem 3.22 from  Haussler et al. [4]) holds. This implies that our regret bound of $\mathcal{O}(\sqrt{\theta T \log K})$ for the FTRL algorithm is optimal in both $T$ and $K$, and note we have $\theta \leq M^2$ (under Assumption 2.1).
>
> **(Q2) Regarding the plots**. The $Y$-axis in the plots represents the mean ($\pm$std) cumulative loss over 40 repetitions of the experiments. In this case, by definition of the losses (see Appendix F), they are equivalent to the regrets.
>
> We will add more clarifications in the next revision. Thank you again for your thoughtful feedback.
>
> **References**
>
> [1] Vovk, V. (2001). Competitive on‐line statistics. International Statistical Review, 69(2), 213-248.
>
> [2] Lugosi, G., and Mendelson, S. (2019). Mean estimation and regression under heavy-tailed distributions: A survey. Foundations of Computational Mathematics, 19, 1145-1190.
>
> [3] Cesa-Bianchi, N., and Lugosi, G. (2006). Prediction, learning, and games. Cambridge University Press.
>
> [4] Haussler, D., Kivinen, J., and Warmuth, M. K. (1998). Sequential prediction of individual sequences under general loss functions. IEEE Transactions on Information Theory, 44(5), 1906-1925.
>
> [5] Zimmert, J., and Seldin, Y. (2021). Tsallis-inf: An optimal algorithm for stochastic and adversarial bandits. Journal of Machine Learning Research, 22(28), 1-49.
>
> [6] Amir, I., Attias, I., Koren, T., Mansour, Y., and Livni, R. (2020). Prediction with corrupted expert advice. Advances in Neural Information Processing Systems, 33, 14315-14325.

---

> > ### Comment · Reviewer_XFbG · 2025-08-01
> >
> > I thank the authors for their response. My concerns have been addressed and I maintain my original score.

---

> > > ### Author Response · Authors · 2025-08-07
> > >
> > > We are happy that your concerns have been addressed. Thank you again for your time and engagement!

---

### Official Review · Reviewer_ke8c · 2025-07-01

**Clarity:** 2
**Significance:** 3
**Originality:** 2
**Rating:** 5
**Confidence:** 3

**Summary:**

This paper studies the experts problem when the losses are "heavy tailed", i.e., when the second moment of the losses are bounded above by an (unknown) constant $ \theta $. The authors show that existing algorithms have lower-order terms that can make the regret as large $ \sqrt{K T} $. The authors develop algorithms whose regret is at most $ O (\sqrt{ \theta T \log (K)}) $. They also show improved results under more restricted settings. Their algorithms are online mirror descent and follow-the-regularized-leader, using clipped versions of instantaneous regret as the loss and using empirical variance-based entropy regularization.

**Questions:**

1. When you discuss Algorithm 2, Line 283 says that "Note that, differently from Algorithm 1, we replace the multi-scale entropic regularizer $\psi_t $ with its Bregman divergence." This is not clear from the main text because Algorithm 2 is only presented in the appendix. Could you elaborate on this point? I looked at the algorithm in the appendix and did not quite understand what you mean.

2. Is it fair to say that choosing $ \eta_{t,i} $ as a function of the empirical variance and second moment of the instantaneous regret is the key to obtaining your result and constitutes the core novelty of your contribution? I can see that the definition of $ \eta_{t,i} $ helps algebraically in the equation after Line 266. However, can you provide some more intuition behind the definition of $ \eta_{t,i} $?

**Ethical Concerns:**

["NO or VERY MINOR ethics concerns only"]

**Final Justification:**

I am satisfied with the authors' response and I have raised my score. I would like the authors to address the issues around clarity and algorithm presentation in the final version. I would also encourage them to include part of their answer to Q2 in their paper - it might be helpful for readers!

**Limitations:**

Yes.

**Quality:**

3

**Strengths And Weaknesses:**

Strengths:
* The authors study an important problem, namely, achieving sublinear regret in the experts problem under heavy-tailed losses, which can occur in many real-world scenarios.
* The core novelty of their contributions is an adaptation of the multi-scale entropy regularization of Bubeck et al. (2019): the authors use time-varying (and expert-varying as in Bubeck et al.) scales ( $ \eta_{t,i} $ ), set the scale to a function of the observed variance and second moment of the instantaneous regret, and perform clipping per expert.
* Section 4 provides concrete examples showing lower-order terms can be large and the authors do so by using a simple construction.

Weaknesses:
* My main concern about the paper is its clarity. While the paper is well written overall, the writing is a bit sloppy in some places.
    * Lines 47-56 and the sentences following Theorems 2.2 and 2.4 seem to have the algorithms switched. For instance, line 47 says that Algorithm 1 achieves $ O (\sqrt{ \theta T \log (K)}) $ worst-case regret. However, Line 78 (following Theorem 2.2) says this bound is achieved by Algorithm 2.
    * The abstract says that logarithmic regret is achieved under a separation assumption and i.i.d. losses. However, the main text and the appendix do not make the assumption of i.i.d. losses to achieve logarithmic regret.
    * The authors refer to three algorithms in the main text for their regret bound results. However, they only present one of them in the main text. It is hard to follow their discussion that references the algorithm design when they don't present those algorithms in the main text.
    * Related to the previous point, in Section 5.2 when the authors discuss Algorithm 2, Line 283 says that "Note that, differently from Algorithm 1, we replace the multi-scale entropic regularizer $\psi_t $ with its Bregman divergence." This is not clear from the main text because Algorithm 2 is only presented in the appendix. Question to the authors: Could you elaborate on this point? I looked at the algorithm in the appendix and did not quite understand what you mean.
* Another weakness is novelty. The core novelty of the contribution, described above, is also a bit of a weakness: the authors combine existing ideas of clipping and multi-scale entropy regularization. (The authors do make the scale data dependent by using empirical second moments.)

---

> ### Author Rebuttal · Authors · 2025-07-30
>
> We sincerely thank you for your detailed and constructive feedback. We are glad you found the problem important and our analysis insightful. We address the points you raised below.
>
> **(W.1.1, W.1.3) Regarding the typos and algorithm presentation**. Thank you for pointing out unclear points and typos. We will fix these in the next version and take advantage of the additional content page to include the algorithms in the main body to improve readability.
>
> **(W.1.2) Regarding the i.i.d. assumption in the abstract**. Our use of "i.i.d. losses" in the abstract was intended as a simple, concrete example of a scenario satisfying our more general assumption Assumption 2.3 (self-bounded environment). If losses are i.i.d. and there is a unique best expert $i^\star$, the gap to any other expert $i$, $\Delta_i$, is positive and lower-bounded by $\Delta_{\min}$. This implies Assumption 2.3 with a constant $C = 0$, since the regret satisfies
> $$R_T = \mathbb{E} \left[ \sum_{t=1}^T \sum_{i \neq i^\star} p_t(i) \Delta_i \right] \geq \mathbb{E} \left[ \sum_{t=1}^T \sum_{i \neq i^\star} p_t(i) \Delta_{\min} \right] = \mathbb{E} \left[ \sum_{t=1}^T (1 - p_t(i^\star)) \Delta_{\min} \right].$$
> We will clarify that Assumption 2.3 is more general and covers non-i.i.d. settings (see, e.g., [1, 2]). See also our answer to Reviewer XFbG for a related discussion.
>
> **(W.1.4, Q.1) Regarding the regularizer in Algorithm 2**. The reason for using the Bregman divergence $\varphi_t(\cdot) = D_t(\cdot \| p_1)$ rather than the multi-scale entropy $\psi_t$ for Algorithm 2 is essentially because we wanted a simplification in the analysis. The Bregman divergence is non-negative, which allows us to simplify some steps in the analysis that involve the decreasing learning rates $\eta_{t,i}$. As we comment at lines 282-290, using the Bregman divergence also guarantees monotonicity over $t$, which makes the FTRL analysis simpler by allowing us to remove the typical sum $\sum_t(\varphi_{t}(p_{t+1})-\varphi_{t+1}(p_{t+1})) \le 0$ from the stability term. On the other hand, monotonicity is not guaranteed for the multi-scale entropic regularizer $\psi_t$, and thus one would need to carefully analyze the corresponding sum $\sum_t(\psi_{t}(p_{t+1})-\psi_{t+1}(p_{t+1}))$ from the stability term of FTRL. We think that the analysis, although slightly more involved, could also lead to a meaningful result in our setting. However, due to the fact that the learning rates we employ depend on $i$ too, this could require the algorithm to tune the learning rates $\eta_{t,i}$ in a slightly different way than currently done by Algorithm 2.
>
> **(W.2, Q.2) Regarding the novelty**. While the ideas we use appear in some form or another in the literature, it is important to note that, as they appear in previous work, they still lead to the lower-order term that can force the regret bound to scale as $\sqrt{K T}$. For example, applying the clipping as proposed by Cutkosky [3] or Orseau and Hutter [4] also leads to the lower-order terms (see lines 126-135). Only when we overhaul these ideas and carefully combine them are we able to avoid the problematic lower-order term.
>
> **(Q2) On the choice of the learning rates $\eta_{t, i}$**. Central to our results is not just the choice of the learning rates $\eta_{t, i}$ but also the clipping we use. At a high level, the choice of learning rate $\eta_{t, i}$ is driven by the need for stability in the analysis. For entropy-regularized OMD/FTRL, the term in the exponent must remain small, e.g. $|\eta_{t, i} \ell_t(i)| \leq 1$, to ensure the quadratic $1 - x + x^2$ is a good approximation of the exponential $\exp(-x)$, where we take $x = \eta_{t, i} \ell_t(i)$ (see, e.g., Section 11.4 of [5] for an argument specific to the Exponential Weights algorithm with rewards, which can be similarly extended to losses and entropy-regularized OMD/FTRL). A standard way to achieve this is to choose a global learning rate $\eta_t \propto 1 / \max_{s \leq t, j} |\ell_s(j)|$. However, this is too conservative: a single large loss from one expert makes the learning rate small for *all* experts, causing the problematic $\sqrt{K T}$ regret. This justifies setting the learning rate for each expert individually, based on their own observed history. We refer to Appendix A.3 for further details on the caveat of choosing a global learning rate.
>
> The clipping mechanism is then linked to this choice of individual learning rates. For expert $i$ at time $t$, it is set to $1 / \eta_{t, i}$ and acts as follows. If an expert has been stable (i.e., low cumulative variance), its learning rate $\eta_{t, i}$ is large, making the clipping threshold $1 / \eta_{t, i}$ small. The algorithm becomes sensitive and clips even moderate losses, as they would appear as outliers for that expert. Conversely, if an expert has a high cumulative variance, its learning rate $\eta_{t, i}$ is small, making the threshold $1 / \eta_{t, i}$ large and the algorithm will tolerate larger losses since they are expected.
>
> We will revise the paper to incorporate these clarifications. Thank you again for your thoughtful review.
>
> **References**
>
> [1] Zimmert, J., and Seldin, Y. (2021). Tsallis-inf: An optimal algorithm for stochastic and adversarial bandits. Journal of Machine Learning Research, 22(28), 1-49.
>
> [2] Amir, I., Attias, I., Koren, T., Mansour, Y., and Livni, R. (2020). Prediction with corrupted expert advice. Advances in Neural Information Processing Systems, 33, 14315-14325.
>
> [3] Cutkosky, A. (2019). Artificial constraints and hints for unbounded online learning. In Conference on Learning Theory (pp. 874-894). PMLR.
>
> [4] Orseau, L., and Hutter, M. (2021). Isotuning with applications to scale-free online learning. arXiv preprint arXiv:2112.14586.
>
> [5] Lattimore, T., and Szepesvári, C. (2020). Bandit algorithms. Cambridge University Press.

---

> > ### Comment · Reviewer_ke8c · 2025-08-05
> >
> > Thank you for your response. I would encourage you to address the issues around clarity and algorithm in the final version. I would also encourage you to incorporate your answer to Q2 in the paper - it might be helpful to readers!

---

> > > ### Author Response · Authors · 2025-08-07
> > >
> > > We will make those changes for the final version. Thank you for the positive feedback and for your time!

---

### Official Review · Reviewer_fiH4 · 2025-07-03

**Clarity:** 3
**Significance:** 3
**Originality:** 3
**Rating:** 4
**Confidence:** 4

**Summary:**

This paper studied online learning with expert advice with $K$ experts and $T$ days. The simplest form of the problem is for the loss function to be bounded by $[0,1]$ so that we could obtain $O(\sqrt{KT})$ worst-case optimal regret. This paper considered the more general setting of general loss functions with “adaptive” losses, i.e., regret bounds in the form of the instance-dependent parameters. On this front, if we know $\max_{t,i}\ell^{t}(i)\leq M$, it is possible to obtain $O(M\sqrt{T \log K})$ regret. This paper instead asked about what will happen if we instead know the second-order moment, i.e., $\mathbb{E}[(\ell^{t}(i))^2]\leq \theta$. Here, the algorithm does not necessarily know the value of $\theta$ in advance.

The paper started by showing an example when $\theta=O(1)$ but algorithm that adaptive to $\max_{t,i}\ell^{t}(i)$ could only result in $\Omega(\sqrt{KT})$ regret. Then, the paper designed an algorithm that achieves $O(\sqrt{\theta T \log{K}})$ regret when $\mathbb{E}[(\ell^{t}(i))^2]$ is bounded by $\theta$. Additionally. For instances that satisfy a “self-bounded environment” assumption, the paper designed an “instance-sensitive” type of algorithm that achieves $O(\theta \log(KT)/\Delta_{\text{min}})$ regret. Finally, for the square loss with the expert losses bounded by $Y$ and second moment bounded by $\sigma$, the paper achieved $O(\log{K}\cdot (Y^2 + \sigma))$ regret.

The main technique of the paper focuses on removing the $M_T = \mathbb{E}[max_{t,i}|r_t|]= \mathbb{E}[max_{t,i}|p_t^{T} \ell^{t} - \ell^{t}(i)|]$. The paper has shown that almost all existing algorithms would lead to this additive term when only the guarantees on the second-order moments are provided. To avoid this term, the main idea of the algorithm appears to be the adoption of the multi-scale entropic regularizer (BDHN [JMLR’19]) and the clipping of the loss when the variance of the algorithm is high. The paper then showed that the additional cost induced by cost clipping is tolerable (Equation (3)). Due to time constraints and my experience with the literature, I did not check the proof.

**Questions:**

I do not have additional questions. Do the authors have responses to the weaknesses I flagged?

**Ethical Concerns:**

["NO or VERY MINOR ethics concerns only"]

**Final Justification:**

====== Post Rebuttal ======
The author pointed out that their paper does contain experiments, which I missed during the initial review. I looked into that section and found that although the settings are limited, experimental results have demonstrated the better performance of their algorithms in the heavy-tail settings.

I've increased my confidence score to reflect this discussion. However, since I'm not very familiar with the existing techniques, I still cannot give a higher score to "champion" for the paper.

**Limitations:**

N/A, this is a theoretical work about online learning, and I do not see any immediate societal impacts.

**Quality:**

3

**Strengths And Weaknesses:**

In my opinion, the paper is generally well-written, and the main conceptual message is clearly articulated: in the bandit setting, one cannot adapt to the parameter $\theta$, while in the full-information expert setting, we could achieve $O(\sqrt{\theta T \log{K}})$ regret without knowing $\theta$ a priori. The bounds in related work were also thoroughly discussed.

The technical part of the paper was written reasonably well. I’m not very familiar with the adaptive algorithm in the expert/bandit settings, but I could nevertheless follow the argument used in section 4. The techniques become harder to follow in section 5, partly due to the fact that I’m not very familiar with the existing techniques in the literature (for adaptive regrets).
The proof sketch gave a nice high-level framework, but the parameters are still quite involved.

On the flip side, I think the paper could have done a better job of motivating the setting from a *practical* perspective. From the paper, it is not very clear in what application scenarios we would have bounded second-order moments. I’m OK with this for a mostly theoretical paper, but for this conference, having better practical motivations will definitely be more helpful.

The above point is also related to the significance of the results. I personally believe the separation between the expert vs. bandit case is of good conceptual significance. However, since I’m not very familiar with the literature, I’ll see what my colleagues have to say about the significance.

The paper does not contain experiments. Again, I'm personally OK with this since this is a theoretical work. However, I do want to flag this for the AC in case that is part of their decision.

Overall, I will be happy to see the paper get accepted, but I cannot give a firm verdict about the significance and the practical aspect. I also did not get a time to carefully check the proof, so I can only give a weak supportive opinion.

====== Post Rebuttal ======
The author pointed out that their paper does contain experiments, which I missed during the initial review. I looked into that section and found that although the settings are limited, experimental results have demonstrated the better performance of their algorithms in the heavy-tail settings.

I've increased my confidence score to reflect this discussion. However, since I'm not very familiar with the existing techniques, I still cannot give a higher score to "champion" for the paper.

---

> ### Author Rebuttal · Authors · 2025-07-30
>
> Thank you for taking the time to review our paper and for your positive feedback. We are pleased you found the paper well-written and conceptually relevant. We address the flagged weaknesses below.
>
> **Regarding the experimental evaluation**. We would like to clarify that we did include experiments in Appendix F (pages 32-34). These experiments empirically validate our theoretical claims by comparing our proposed algorithms (LoOT-Free OMD and FTRL) against several established baselines on synthetic heavy-tailed data. The results, shown in Figures 1 and 2, illustrate the failure modes of existing methods that our work addresses, confirming the practical implications of our analysis.
>
> **Regarding practical motivation**. Thank you for this suggestion. While our work is primarily theoretical, the assumption of heavy-tailed losses is motivated by several real-world scenarios where data is prone to infrequent, large-magnitude events. For instance:
>
> - _Financial markets_: Asset returns are widely recognized as being heavy-tailed (see, e.g., [1]). Our framework can be applied to problems like online portfolio selection, where an investor aggregates the advice of different financial experts or trading strategies, and the daily losses (or negative returns) can exhibit large fluctuations.
>
> - _Energy forecasting_: Forecasting electricity prices or demand is another key application. For instance, Li and Jones [2] observe that time series often feature sudden, large spikes due to unpredictable events (e.g., power plant outages, extreme weather). Aggregating different forecasting models as experts is a common approach in this domain [3].
>
> - _Online media and e-commerce_: The popularity of items like news articles, videos, or products often follows a power-law distribution. In applications like ad placement or content recommendation, where the goal is to predict engagement, the observed outcomes (e.g., clicks, views, sales) can be heavy-tailed (see, e.g., [4]).
>
> We will incorporate these examples in the final version of the paper to better motivate our contributions. We hope these clarifications address your concerns and are grateful for your supportive assessment.
>
> **References**
>
> [1] Bradley, B. O., and Taqqu, M. S. (2003). Financial risk and heavy tails. In Handbook of heavy tailed distributions in finance (pp. 35-103). North-Holland.
>
> [2] Li, Y., and Jones, B. (2019). The use of extreme value theory for forecasting long-term substation maximum electricity demand. IEEE Transactions on Power Systems, 35(1), 128-139.
>
> [3] Devaine, M., Gaillard, P., Goude, Y., and Stoltz, G. (2013). Forecasting electricity consumption by aggregating specialized experts: A review of the sequential aggregation of specialized experts, with an application to Slovakian and French country-wide one-day-ahead (half-) hourly predictions. Machine Learning, 90(2), 231-260.
>
> [4] Cha, M., Kwak, H., Rodriguez, P., Ahn, Y. Y., and Moon, S. (2007, October). I tube, you tube, everybody tubes: analyzing the world's largest user generated content video system. In Proceedings of the 7th ACM SIGCOMM Conference on Internet measurement (pp. 1-14).

---

> > ### Comment · Reviewer_fiH4 · 2025-08-05
> >
> > > We would like to clarify that we did include experiments in Appendix F (pages 32-34). These experiments empirically validate our theoretical claims by comparing our proposed algorithms (LoOT-Free OMD and FTRL) against several established baselines on synthetic heavy-tailed data. The results, shown in Figures 1 and 2, illustrate the failure modes of existing methods that our work addresses, confirming the practical implications of our analysis.
> >
> > Thanks for pointing this out! My bad, I missed that section in the appendix. The experiments appear good for your algorithms.
> >
> > As I said in my original assessment, I think the paper is generally well-written, and I'm leaning towards acceptance. I'm keeping my score to maintain this opinion. I'll update my review to reflect this discussion.

---

> > > ### Author Response · Authors · 2025-08-07
> > >
> > > Thank you for the positive evaluation and feedback! We appreciate your engagement.

---

### Official Review · Reviewer_QACD · 2025-07-07

**Clarity:** 4
**Significance:** 3
**Originality:** 3
**Rating:** 5
**Confidence:** 4

**Summary:**

This paper provides algorithms for prediction with expert advice with heavy tailed losses (for the conditional distribution of each instance given the history). It correctly points out a critical flaw in applying standard scale-adaptive methods to the case of conditionally heavy tails (that the random variations form sampling will dominate the regret giving $\sqrt{NT}$ bounds instead of $\sqrt{\theta T\log N}$ when the scale^2 is $\theta$). The paper provides new algorithms which depend on second moments rather than expected maxima, and hence are better order. The algorithms are variants of hedge/exponential weights, instantiated as OMD or FTRL respectively, with different learning rates per expert (given by a diagonal weight matrix). The weights are updated in a way similar to other adaptive methods, but using clipped losses to "truncate" the heavy tails.

**Questions:**

-

**Ethical Concerns:**

["NO or VERY MINOR ethics concerns only"]

**Final Justification:**

i am still in favour of accepting. i maintain my score

**Limitations:**

-

**Paper Formatting Concerns:**

-

**Quality:**

3

**Strengths And Weaknesses:**

This paper is well written, clearly identifies and resolves a relevant problem, and provides practical guidance.

---

> ### Author Rebuttal · Authors · 2025-07-30
>
> Thank you for taking the time to review our paper and for the positive feedback. We are glad you found our paper well-written and recognized the relevance of our contributions. We are available to answer any questions that may arise during the discussion period.

---

### Decision · Program_Chairs · 2025-09-17

**Decision:**

Accept (poster)

**Comment:**

This paper studies algorithms for prediction with expert advice with heavy-tailed losses, noting that standard methods fail for conditionally heavy tails due to lower-order terms dominating the overall loss due to large variances from sampling. The paper instead achieves the optimal regret bounds by incorporating multi-scale entropic regularizers and clipping the loss when the variance of the algorithm is high. All reviewers agree that the paper clearly articulates the motivation for studying heavy-tailed losses and that the corresponding algorithmic guarantees given in this paper are valuable.

Nevertheless, reviewer feedback pointed out several points that could further strengthen the potential impact of the paper; I encourage the authors to take these points into consideration.